



# Tropical cirrus evolution in a km-scale model with improved ice microphysics

Blaž Gasparini[1], Rachel Atlas[2], Aiko Voigt[1], Martina Krämer[3,4], and Peter N. Blossey[5]

[1]Department of Meteorology and Geophysics, University of Vienna, Vienna, Austria
[2]CNRS-Laboratoire de Météorologie Dynamique, LMD, Palaiseau, France
[3]Institute for Atmospheric Physics, University of Mainz, Mainz, Germany
[4]IEK-7, Forschungszentrum Jülich, Jülich, Germany
[5]Department of Atmospheric and Climate Science, University of Washington, Seattle, USA

**Correspondence:** Blaž Gasparini (blaz.gasparini@univie.ac.at)

**Abstract.**

Tropical cirrus clouds form via in situ ice nucleation below the homogeneous freezing temperature of water or detrainment from deep convection. Despite their importance, limited understanding of their evolution and formation pathways contributes to large uncertainty in climate projections. To address these challenges, we implement novel passive tracers in the cloud-resolving
model SAM to track the three-dimensional development of cirrus clouds. One tracer tracks air parcels exiting convective updrafts, revealing a rapid decline in ice crystal size and number as anvils age. Another tracer focuses on in situ cirrus, capturing their formation in the cold upper atmosphere and the subsequent reduction in ice crystal number over time. We find that in situ cirrus dominate at colder temperatures and lower ice water contents, while anvil cirrus prevail at temperatures above -60°C. Although in situ cirrus have a smaller radiative impact compared to anvil cirrus, their contribution must be considered when
evaluating top-of-the-atmosphere radiative effects. These findings improve our ability to assess the distinct roles of convective and in situ cirrus in shaping tropical cirrus properties and their impacts on climate.

We also improve the model's representation of tropical cirrus through simple, computationally inexpensive microphysics modifications, achieving better agreement with tropical aircraft observations. We show that updrafts critical for tropical cirrus formation are only resolved at horizontal grid spacings finer than 250 m—much finer than those used in global storm-resolving
models. To mitigate this limitation, we propose microphysics improvements that reduce biases without increasing computational costs.

## 1 Introduction

Tropical cirrus clouds, defined as ice clouds with tops at temperatures colder than -40°C, dominate regions of tropical ascent in both cloud fraction and radiative effects (Berry and Mace, 2014; Hartmann and Berry, 2017). These clouds are diverse,
with their origins and properties shaped by distinct formation mechanisms: convective cirrus originating from deep convective updrafts and in situ cirrus that are formed by ice nucleation in the cold tropical upper troposphere. Understanding the relative contributions and characteristics of these two cloud types is crucial for improving their representation in atmospheric models.



The distinction between convective and in situ cirrus has practical implications for climate projections, as their formation mechanisms may respond differently to greenhouse gas forcing, potentially leading to different cloud feedbacks.

Convective cirrus, or anvil clouds, are initially thick and optically dense but rapidly lose mass through precipitation as they spread horizontally over large areas (Deng et al., 2016). Over time, they evolve into thinner clouds, often with optical depths of 1 to 2, which represent the most common form of tropical cirrus (Sokol and Hartmann, 2020). In contrast, in situ cirrus typically form in the tropical tropopause layer (TTL), above the mean detrainment level of convection. These clouds arise from ice nucleation triggered by small-scale dynamical processes, such as gravity wave-induced fluctuations (Hoyle et al., 2005; Jensen et al., 2013). Unlike anvils, in situ cirrus are optically thin and display distinct microphysical properties, such as very small ice crystals (Krämer et al., 2020).

    This study focuses on the lifecycle and microphysical evolution of both anvil and in situ cirrus. Snapshots or long-term averages of cloud properties, typically provided by model output or observations, do not reveal enough information to fully understand the processes that shape tropical cirrus. Lagrangian methods, such as tracking cloud properties using trajectories, have been widely used in both models (Wernli et al., 2016; Gasparini et al., 2021; Sullivan et al., 2022) and observations (Horner and Gryspeerdt, 2023; Jeggle et al., 2024). Passive tracers, an alternative approach to disentangle the origin and evolution of cirrus clouds, are more flexible in their use, easier to implement, and computationally more efficient.

    Tropical cirrus formation is shaped by processes spanning a wide range of spatial and temporal scales, from microphysical mechanisms such as ice nucleation, deposition, and sublimation to dynamical influences including gravity waves, turbulence, and mesoscale circulations (Corcos et al., 2023; Jensen et al., 2024; Gasparini et al., 2022). However, even the most advanced models struggle to capture these complexities. Anvils originate in deep convective updrafts, which are unresolved in traditional general circulation models (GCMs). Although global storm-resolving models (GSRMs) with kilometer-scale grid spacings capture larger-scale convective dynamics, they fail to resolve the fine-scale dynamics critical to both anvil and in situ cirrus lifecycle (Atlas and Bretherton, 2023; Köhler et al., 2023; Achatz et al., 2024).

Moreover, GSRMs show large variability in simulating the microphysical properties of tropical cirrus (Atlas et al., 2024) and their radiative effects (Turbeville et al., 2022), sometimes performing worse than traditional GCMs. This reflects the trade-offs inherent in GSRMs, where the computational costs of high horizontal resolution are typically offset by simplified parameterizations of subgrid processes, particularly cloud microphysics.

    In this work, we demonstrate that simple and inexpensive modifications to cloud microphysics can largely improve the sim-
ulation of tropical cirrus. By tracking cloudy air parcels from detrainment or in situ nucleation, we identify key differences in their lifecycle and microphysical properties, offering new insights into their respective contributions to tropical cirrus climatology and their radiative impacts. Furthermore, while kilometer-scale GSRMs can resolve updrafts near deep convection, we show that the dynamics critical for cirrus cloud formation in non-convective regions are only resolved at hectometer-scale grid spacings. These findings highlight some limitations of current modeling approaches and provide a pathway toward a more
accurate representation of tropical cirrus in climate models.





## 2 Methods

### 2.1 Model

We use the System for Atmospheric Modeling (SAM) cloud resolving model (Khairoutdinov and Randall, 2003) version 6.10.9. SAM uses a 1.5-order Smagorinsky-type closure scheme to represent subgrid-scale turbulence and subgrid-scale motions. The timestep in SAM is adaptive and set based on the Courant-Friedrich-Levy criterion, that typically leads to a timestep of about 4.5 s for simulations at the horizontal grid spacing of 1 km. Radiative fluxes and heating rates are computed with RRTMG (Mlawer et al., 1997; Iacono et al., 2008), which is called every three minutes. Cloud and precipitation processes use the Predicted Particle Property (P3, Morrison and Milbrandt, 2015) microphysical scheme version 3.1.14.

#### 2.1.1 Description of ice nucleation in the standard P3 scheme

In mixed-phase conditions, ice crystals are formed by the following processes (Fig. 1 a):

- *Immersion freezing of cloud droplets and rain*: a volume-dependent formulation from Bigg (1953) with parameters following Barklie and Gokhale (1959).

- *Deposition nucleation*: a temperature-dependent formulation by Cooper (1986) that is limited to relative humidities with respect to ice ($RH_{ice}$) of $> 105\%$ and temperatures colder than -15° C. Alternatively, a supersaturation-dependent Meyers et al. (1992) parameterization can be used. The maximum number of newly nucleated ice crystals is for both deposition freezing mechanisms limited to 0.1 cm$^{-3}$ s$^{-1}$.

- *Homogeneous freezing of cloud droplets and rain* which occurs instantaneously at a temperature of -40°C.

At temperatures colder than -40°C, ice crystals continue to be nucleated by the Cooper (1986) or Meyers et al. (1992) nucleation. In reality, nucleation events at such cold temperatures always lead to ice crystal concentrations of 0.1 cm$^{-3}$ s$^{-1}$, as set by the ice nucleation limit. Similar approaches to ice nucleation are used in a large number of microphysical schemes beyond the one used here (e.g. Morrison et al., 2005; Thompson et al., 2008).

#### 2.1.2 Modifications to ice nucleation

Deposition nucleation parameterizations by Cooper (1986) and Meyers et al. (1992) are based on data from mixed-phase regime and thus should not be active at temperatures colder than the homogeneous freezing temperature of water. We thus limit the deposition nucleation parameterizations to temperatures warmer than -37°C. Moreover, because deposition freezing is thought to be negligible in mixed-phase clouds (e.g. Ansmann et al., 2008; DeMott et al., 2010; Hoose et al., 2008; Lohmann et al., 2016), the ice crystals are allowed to form only in the presence of cloud droplets, effectively changing the deposition freezing parameterizations into a type of condensation freezing. The maximum number of nucleated ice crystals by the modified Cooper (1986) (or, alternatively, Meyers et al., 1992) scheme is increased to 0.15 cm$^{-3}$ s$^{-1}$. While these modifications aim to provide a more physically consistent representation of mixed-phase clouds, we did not explicitly evaluate the scheme's performance



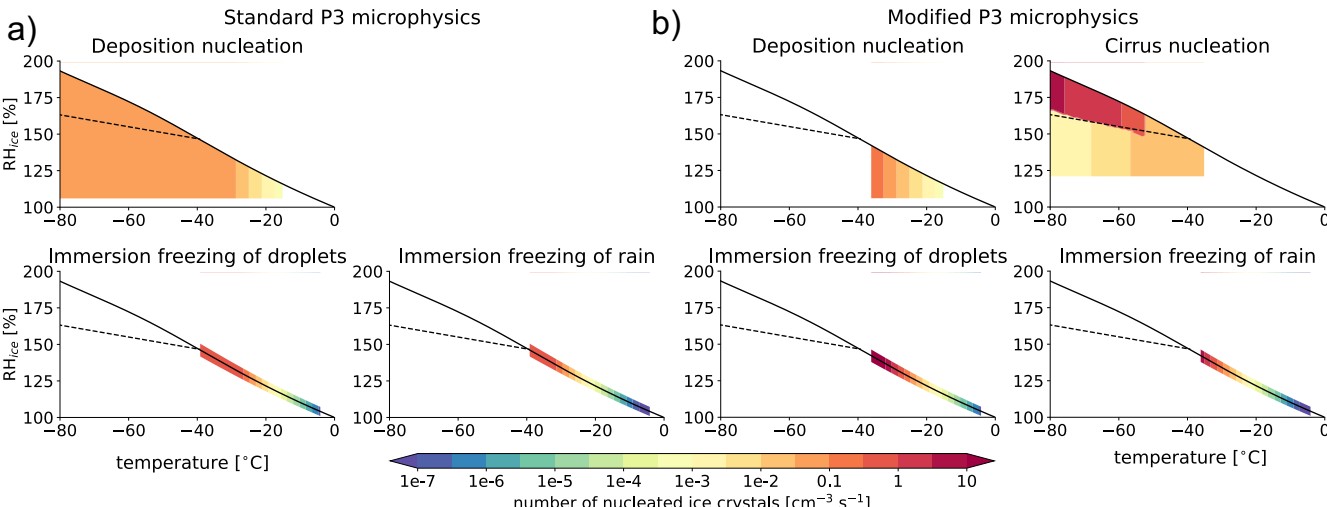

**Figure 1.** A visualization of the number of nucleated ice crystals from ice nucleating schemes in (a) the standard P3 freezing scheme and (b) its modified version.

for this cloud type, as it lies outside the main scope of this study. A number of further refinements to would be necessary to achieve a more accurate simulation of mixed-phase clouds, as discussed later in section 5.1.

As we limited the existing ice nucleation mechanisms to the mixed-phase regime, we implement a new ice nucleation scheme for T<-37°C, which also helps mitigate the bias in ice crystal number concentration (ICNC; see Sec. 5.1 for more detail). The newly implemented approach follows Shi et al. (2015) (Fig. 1b, labeled as cirrus nucleation) and represents the competition between homogeneous and heterogeneous nucleation in cirrus clouds (Liu and Penner, 2005) and the effect of pre-existing ice crystals. The scheme is fed by a predefined, temperature-dependent value of ice nucleating particles (INPs). In this work, the INP number is set to $2 \cdot 10^{-3}$ cm$^{-3}$ at temperatures colder than -70°C and increases linearly to $20 \cdot 10^{-3}$ cm$^{-3}$ at temperatures of -40°C and warmer. While the INP number in the upper troposphere is subjected to large uncertainties, the suggested numbers are plausible for relatively clean, aerosol-free environments in the tropical Pacific and follow model-simulated INP concentrations (Gasparini and Lohmann, 2016). The number of sulfate aerosols is set to 20 cm$^{-3}$ and is thus not a limiting factor in ice nucleation.

Very importantly, the maximum allowed ICNC is relaxed from a very limiting 0.5 to 20 cm$^{-3}$. Such high concentrations were occasionally observed in aircraft measurements of fresh anvils (Krämer et al., 2020; Jensen et al., 2018). Increasing the ICNC limit alone was previously shown to change anvil cloud properties, leading to more thin cirrus (e.g., Fig. 11a in Gasparini et al., 2019). The cirrus nucleation scheme requires the input of an updraft velocity, and we choose to input the sum of the resolved vertical wind and an estimate of subgrid-scale updraft strength derived from the subgrid-scale turbulent kinetic energy ($TKE_{SGS}$) as $W_{TKE,SGS} = \sqrt{0.667 \cdot TKE_{SGS}}$, given that not all updrafts relevant for cloud formation are resolved at horizontal grid spacings of about 1 km (see more in Sec. 4.3). The $W_{TKE,SGS}$ term is computed assuming that




the $TKE_{SGS}$ ($TKE_{SGS} = 1/2 \cdot (u'u' + v'v' + w'w')$) is equally partitioned in the three directions ($w'w' = u'u' = v'v'$), and therefore $TKE_{SGS} = 3/2 \cdot w'w'$, where $W_{TKE,SGS} \equiv w'$.

Finally, we use a more accurate formulation for the saturation vapor pressure of liquid water and ice (Murphy and Koop, 2005), replacing the Flatau et al. (1992) formulation, which performs poorly at cold temperatures in the TTL. This change particularly affects in situ ice nucleation at temperatures below -70°C in the TTL (see Fig. 11 in Murphy and Koop, 2005).

Additionally, we adjust the homogeneous freezing threshold for cloud droplets from -40°C to -37°C. While droplets can freeze over a wide temperature range, larger droplets may freeze as warm as -35°C (Ickes et al., 2015; Shardt et al., 2022). Despite issues with using a fixed temperature freezing threshold (Herbert et al., 2015), -37°C is a more physically justified value than the -40°C used in the reference version of the P3 microphysical scheme.

### 2.1.3   Passive tracers

We implemented two passive tracers to facilitate the analysis of the tropical cirrus lifecycle. The "time after detrainment" or simply "detrainment" tracer, denoted $A$, evolves as:

$$A(\mathbf{x}, t) = 1 \quad \text{where } |w| > 1 \frac{m}{s}, \ q_c + q_i > 10^{-6} \frac{kg}{kg}, \text{ and } T_\rho' > 0 \tag{1}$$

$$\frac{\partial A}{\partial t} = -\frac{A}{\tau_A} \quad \text{elsewhere} \tag{2}$$

where $w$ is the vertical velocity, $q_c$ and $q_i$ the cloud liquid and ice mass mixing ratios, $T_\rho'$ the density temperature anomaly from

the domain mean (which is proportional to buoyancy), and $\tau_A = 80$ minutes is an arbitrary decay timescale for $A$. Neglecting the effects of subgrid-scale mixing on the passive tracer, $A$, the time since detrainment from active convection can be calculated as:

$$\tau_{detr} = -\tau_A \times log(A) \tag{3}$$

The behavior of the tracer in the context of idealized tropical convection is described in the appendix of Gasparini et al. (2022).

Additionally, we implement an analogous tracer to determine the time after in situ ice nucleation. The tracer is set to 1 in all grid cells with active cirrus ice nucleation and decays elsewhere with the same decay timescale $\tau_A$. We note that both implemented tracers follow air parcels and not ice crystals, which results in biases when ice crystals sediment out of air parcels.

### 2.2   Simulation setup

We use a tropical channel setup that is wide in the zonal direction (3888 km) and narrow in the meridional direction (36 km)

with double-periodic boundary conditions. The prescribed sea surface temperatures vary sinusoidally in the zonal direction between 24°C at the domain edge and 28°C in the middle of the domain. Convection develops over the warmer SSTs and gives rise to a large-scale overturning circulation in the zonal direction reminiscent of the Walker circulation. Such a "mock-Walker" circulation setup is therefore appropriate for studying the interplay between convection, clouds, and radiation in the tropics (Bretherton et al., 2006; Bretherton, 2007; Wing et al., 2023; Silvers et al., 2023). To accurately represent processes in





the TTL, we impose a mean large-scale vertical velocity based on observations by Yang et al. (2008). Zonal winds increase linearly with altitude from 0 m s$^{-1}$ at surface to 5 m s$^{-1}$ at altitudes above 14 km. Our "mock-Walker" setup follows the one described in more detail by Blossey et al. (2010).

### 2.3    In-situ observations of ice cloud properties and updraft velocities

For a fairer comparison with model data, which simulates a climate comparable to that of the tropical Pacific, we use airborne
data from three campaigns in the tropical western Pacific: Airborne Tropical TRopopause EXperiment (ATTREX; Jensen et al., 2017); Pacific Oxidants, Sulfur, Ice, Dehydration, and cONvection experiment (POSIDON, Jensen et al., 2018); and the CONvective TRansport of Active Species in the Tropics (CONTRAST) Experiment (Pan et al., 2017).

Vertical velocity data are only used from the tropical western Pacific flights of the ATTREX and POSIDON campaigns. The data are sampled using NASA's Meteorological Measurement System (MMS) instrument (Scott et al., 1990), which has a time
resolution of 20 Hz. The vertical velocity variance is computed at 1 Hz after the data has been corrected by detrending and removing the mean of each flight leg. More details on the processing of updraft velocities are described in Atlas and Bretherton (2023).

In situ data of ice cloud properties for the three campaigns is taken from the Krämer et al. (2020) dataset. The data from the POSIDON and ATTREX field campaigns was recently corrected for a bug in the estimate of ICNC, which substantialy
increased the ICNC. The measurement resolution is 1 Hz, with an aircraft velocity of 170 m/s in POSIDON and ATTREX and 200 m/s in the CONTRAST campaign. The lower limit of detectable particle concentration in a given sampling time depends on the aircraft velocity and sampling area of the respective instrument (Krämer et al., 2020; Costa et al., 2017). This limits the measurements for CONTRAST of ice crystals at the ICNC <0.01 cm$^{-3}$ and mean mass ice radii smaller than 35 $\mu$m. For consistency, model output under such conditions is not included in the calculation of model performance. Due to the slower
aircraft speed and different instrumentation, there is no such limitation for POSIDON and ATTREX. The dataset contains only few measurements within or very near active deep convection.

### 2.4    Satellite retrievals

We use satellite retrievals from the years 2007-2010 for the tropical western Pacific (TWP) (20°S-20°N, 145°-180E°), an almost exclusively ocean-covered area characterized by persistent deep convection throughout the year.

### 2.4.1    DARDAR

This dataset is derived from combined retrievals by the CloudSat Cloud Profiling Radar (Stephens et al., 2008) and the Cloud-Aerosol Lidar with Orthogonal Polarization (CALIOP, Winker et al., 2010). Merged CloudSat radar reflectivity and CALIOP lidar attenuated backscatter signals were used to build the radar/lidar product (DARDAR, Delanoë and Hogan, 2008, 2010) that retrieves ice water content (IWC), effective ice crystal size, and the extinction coefficient. DARDAR has a horizontal footprint
of 1.7 km and a vertical resolution of 60 m. We only use the vertically integrated IWC, which includes all frozen hydrometeors





and is denoted here as ice water path (IWP). Since the lidar signal is noisier during daytime, resulting in the detection of fewer thin clouds (Avery et al., 2012), we use nighttime-only data. As one measure of uncertainty in the retrievals, IWP comparisons are made with both the newest dataset (DARDARv3) and an older version of the retrieval algorithm (DARDARv2) (Cazenave et al., 2019).

**2.4.2 2C-ICE**

We also use retrievals of IWP from the Cloudsat and CALIPSO Ice Cloud Property Product (2C-ICE) version RF05 (Deng et al., 2015). Despite originating from the same input data as the DARDAR product, its ice properties are derived using different assumptions and therefore helps quantify the uncertainty in satellite-retrieved quantities through comparisons with DARDAR nighttime-only data.

**2.5 Cloudsat-Calipso-CERES-MODIS (CCCM)**

The CALIPSO-CloudSat-CERES-MODIS (CCCM) dataset (Kato et al., 2011) merges cloud fraction data from CALIPSO lidar (Winker et al., 2010) and CloudSat radar (Stephens et al., 2008) with MODIS IWP data and CERES radiative fluxes (Wielicki et al., 1996). CCCM's horizontal resolution is about 30 km, equivalent to CERES retrievals. Shortwave (SW) radiative fluxes from CERES used in this work are from the measured fluxes during the 1:30 pm satellite overpass, accounting for diurnally

averaged insolation values. Data points with zenith angles greater than 70° are excluded to mitigate issues at high solar zenith angles. Albedo is computed based on incoming and outgoing SW fluxes at the top of the atmosphere (TOA). The average reflected SW flux during the day is calculated by multiplying the albedo by the daily and yearly average incoming radiation, set at 409.6 W m$^{-2}$, the annual average insolation for the band between 20°S and 20°N (Wing et al., 2018). This ensures that values of radiative fluxes are comparable to the climatological cloud radiative effects.

The TOA albedo ($\alpha$) and SW cloud radiative effect (CRE) are computed as

$$\alpha = \frac{SW_{out}}{SW_{in}} \qquad (4)$$

$$SW_{CRE} = -(\alpha - \alpha_{clear-sky}) \times 409.6 W m^{-2} \qquad (5)$$

In addition, Fig. B2 compares model computed quantities with directly retrieved radiative fluxes to ensure better consistency with retrievals. SW CRE is thus computed as

$$SW_{CRE} = -(SW_{out} - SW_{out,clear-sky}) \qquad (6)$$

The LW CRE is computed as

$$LW_{CRE} = OLR_{clear-sky} - OLR \qquad (7)$$

where OLR is outgoing LW radiation at the TOA.



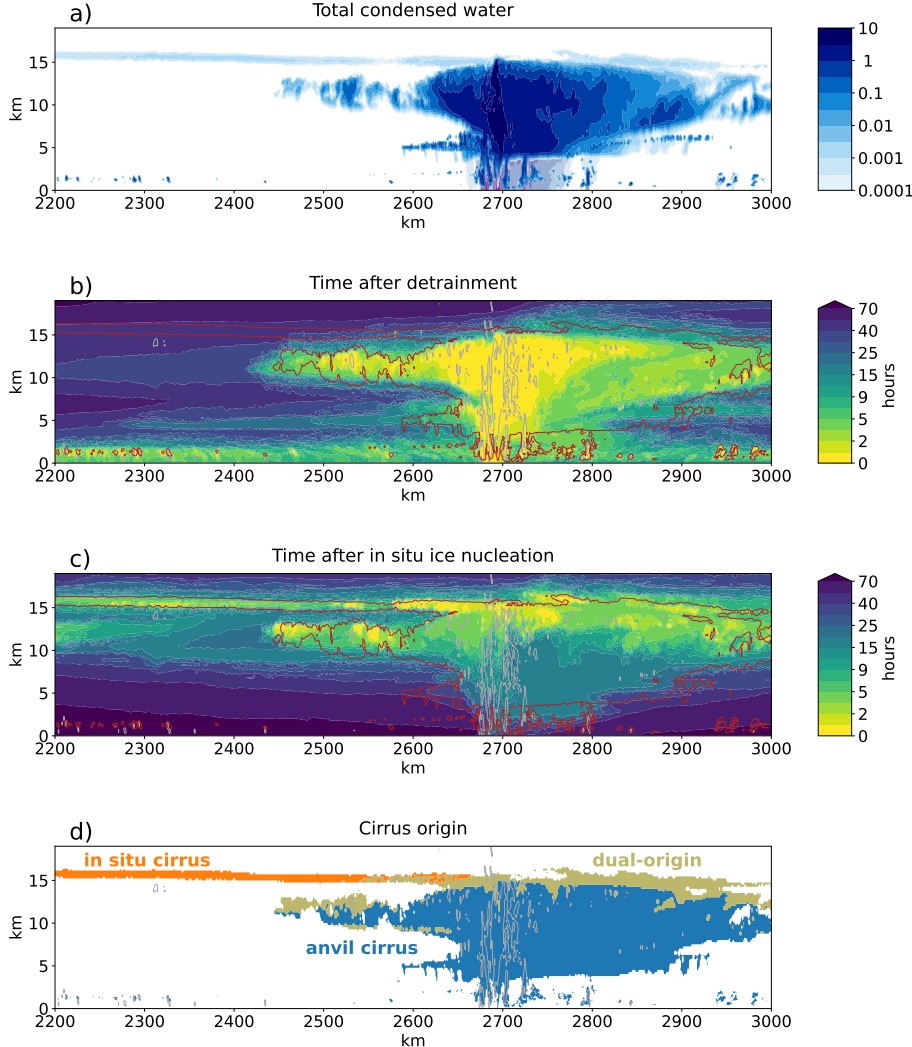

**Figure 2.** (a) A simulated mesoscale convective system with an extensive anvil cloud shield and overlying TTL cirrus. Purple contours in a) indicate rain. Panels b) and c) show values of detrainment and in-situ nucleation tracers for the same cloud system. Panel d) presents the outcome of a cirrus origin classification criterion. In situ cirrus (in orange) are defined as cloudy parcels that have not been in contact with detrained air for at least 30 hours and where the time since in situ nucleation tracer is shorter compared to the time since detrainment. The remaining clouds are classified as anvil cirrus (blue). Portions of the anvil where the time since in situ nucleation tracer is shorter compared to the time since detrainment are classified as "dual-origin" (in brown). Gray contours delineate updraft velocities of $1 \text{ m s}^{-1}$. Red contours in b) and c) delineate total cloud condensate of $1 \cdot 10^{-4} g kg^{-1}$.





# 3 Origin and evolution of cirrus

## 3.1 Cirrus origin

We use a case study and statistical estimates of cirrus origin to demonstrate the utility of passive tracers in disentangling the contributions of convective and in situ processes in the SAM model with the improved microphysical scheme (Sec. 2.1.2).

Figure 2 depicts a snapshot of a multicore mesoscale convective system with an anvil cloud shield extending approximately 500 km. This system includes a fresh, thick precipitating anvil cloud that evolves into a thinner, aged anvil cloud. Above this, a thin cirrus layer spans altitudes of 15–16 km. Identifying the origin of such thin cirrus from a single model output timestep is challenging. Although thin TTL clouds are typically formed in situ (Krämer et al., 2016; Huang and Dinh, 2022), they might also be remnants of TTL-penetrating deep convection.

Passive tracers resolve this uncertainty. The detrainment tracer highlights regions of active deep convection and thick anvil clouds with times since detrainment typically less than ten hours and reveals that the overlying thin cirrus resides predominantly in air undisturbed by convection for at least 30 hours (Fig. 2b). The in situ nucleation tracer confirms this, showing that much of the thin cirrus originates from recent ice nucleation events (within the last 5 hours; Fig. 2c).

Interestingly, the nucleation tracer also indicates that some parts of the anvils experience ongoing ice nucleation. These
events, driven by convective gravity waves or within-anvil updrafts, align with previous observational studies (Jensen et al., 2009; Hartmann et al., 2018; Krämer et al., 2020; Sokol and Hartmann, 2020). However, their broader significance for tropical cirrus remains uncertain (Dinh et al., 2023; Gasparini et al., 2023).

To better characterize the microphysical origin of cirrus, we classify them into three categories: pure in situ, anvil, and dual-origin. Anvil cirrus are defined as clouds where the time since detrainment is shorter than the time since in situ nucleation. Pure
in situ cirrus are those not detrained for at least 30 hours, where the time since nucleation is shorter. The dual-origin category includes anvils clouds that are additionally influenced by in situ nucleation within or at anvil edge or cirrus forming near active convection (Fig. 2d). Notably, while dual-origin cirrus are affected by in situ nucleation, their total ice mass and number remain dominated by convective outflow (not shown).

Our tracer approach confirms previous findings highlighting IWC as a good predictor of cirrus origin (Krämer et al., 2016).
High-IWC tropical cirrus are thought to be of convective origin, while low-IWC cirrus, particularly those at cold temperatures, are more likely of in situ origin (Luebke et al., 2016; Krämer et al., 2016). Using a two-dimensional IWC-temperature space, we find that in situ cirrus dominate only for the coldest, low IWC tropical cirrus (IWC $< 10^{-3}$ g m$^{-3}$, T $<$ -70°C), while high-IWC cirrus and most cirrus at warmer temperatures are of convective origin. In situ contributions range from 10% (T $>$ -50°C) to 50% (T $<$ -70°C), an estimate that is likely a lower bound because it excludes portions of the dual-origin category
that could be considered in situ cirrus. Additionally, we repeat the analysis using ICNC, which leads to less distinct patterns (Fig. 3b). While high ICNC bins are clearly associated with anvil cirrus, in situ cirrus fractions remain steady at ~20–40% for ICNC $< 0.03$ cm$^{-3}$, increasing to over 50% only at temperatures colder than -60°C.

Finally, vertically integrated cirrus origin analysis in Fig. 3c shows that in situ cirrus dominate at IWP $< 0.4$ g m$^{-2}$ and peak at IWP values between 0.1 and 0.5 g m$^{-2}$. At IWP $> 10$ g m$^{-2}$, in situ nucleation is highly unlikely due to limited vapor





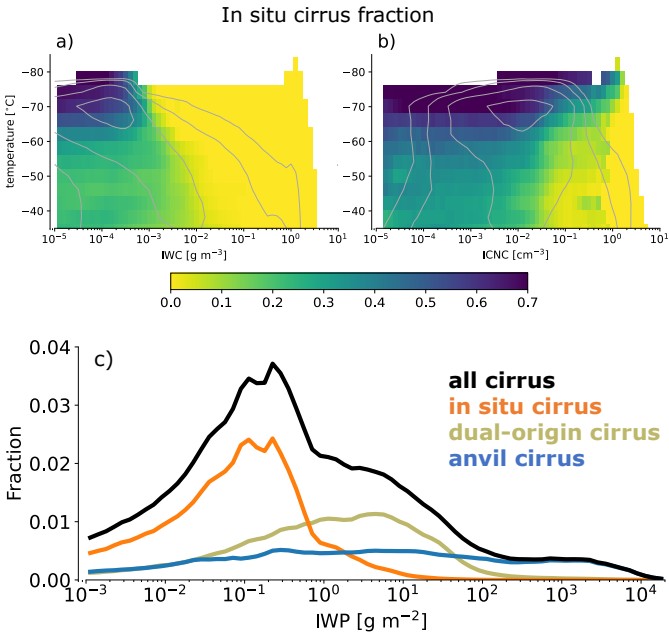

**Figure 3.** Fraction of in situ cirrus represented in the (a) temperature-IWC and (b) temperature-ICNC space for the cirrus classification criterion from Fig. 2. The gray lines in the contour indicate the joint distribution of cloud occurrence in each two-dimensional phase space. Panel c) shows the fraction of cirrus binned by IWP.

availability and slow depositional growth in the TTL. Clouds with IWPs of 1–10 g m$^{-2}$ often consist of contributions from both in situ and convective sources, suggesting that in situ nucleation plays a role in sustaining and prolonging aged cirrus lifetime.

## 3.2 Evolution of tropical cirrus

The two passive tracers provide new insights into the distinct microphysical evolution pathways of detrained anvils and in situ cirrus. The tracer data are presented in the ICNC–ice crystal mean mass radius (from now on: ice number–radius) space (Fig. 4), which provides an intuitive aggregated perspective on ice cloud properties, and has been used already in the analysis of observational and model data (Krämer et al., 2016; Gasparini et al., 2018). This perspective helps differentiate and track the lifecycle of numerous cirrus clouds formed in our simulation, offering a more comprehensive understanding of cirrus evolution compared to a snapshot perspective.

For detrained anvils, deep convection initially injects high concentrations (ICNC > 0.1 cm$^{-3}$) of ice crystals spanning a broad size range, including relatively large particles. The tracers allow for the study of how these crystals evolve over time: within the first 1–3 hours, there is a rapid decrease of both number concentration and size due to sedimentation, marking the most dynamic phase of anvil evolution (compare Fig. 4b and c). Beyond this period, the evolution slows, with ice mass and



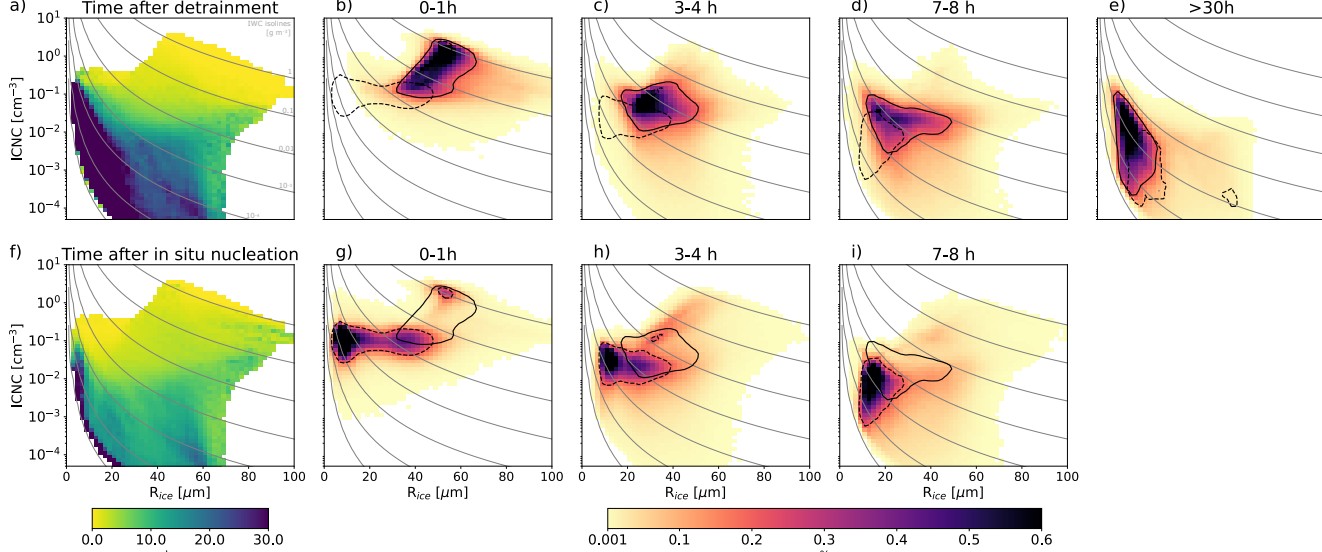

**Figure 4.** Evolution of microphysical properties tracked with passive tracers as a function of ICNC and ice mass radius (computed as the radius of a solid ice sphere with mass IWC/ICNC, as in Krämer et al., 2020). Panels a) and f) represent the mean time of air parcels after detrainment and in situ nucleation. The other panels present the joint distribution of ICNC and mass radius for the stated time after detrainment (upper row) or time after in situ ice nucleation (lower row). The two contour lines encircle the peak probability distribution of particles under the selected conditions (detrainment = solid lines; nucleation = dashed lines). Isolines of IWC are plotted in gray. Since there are very few grid boxes at time after in situ nucleation of more than 30 h, we omit that panel.

number decreasing gradually as sedimentation and sublimation deplete the ice crystals (compare Fig. 4c and d). Panel 4e shows
properties of clouds classified as in situ cirrus in Figs. 2d and 3.

In contrast, freshly nucleated in situ cirrus crystals form at smaller sizes and intermediate concentrations (ICNC: 0.02–
0.2 cm$^{-3}$). Homogeneous nucleation events occasionally spike these concentrations but remain transient and thus are barely visible in our frequency figure. Since most of the in situ crystals are smaller than 30 $\mu$m, sublimation may be a more important ice crystal sink than sedimentation. This may imply a greater sensitivity to atmospheric thermodynamic conditions, such as
temperature and supersaturation fluctuations. Over time, in situ cirrus also lose ice number and size, eventually converging to microphysical properties comparable to those of aged anvil cirrus. In summary, while both in situ and detrained cirrus retain distinct properties in the first 3–5 hours, they become harder to distinguish in the later stages of their evolution.





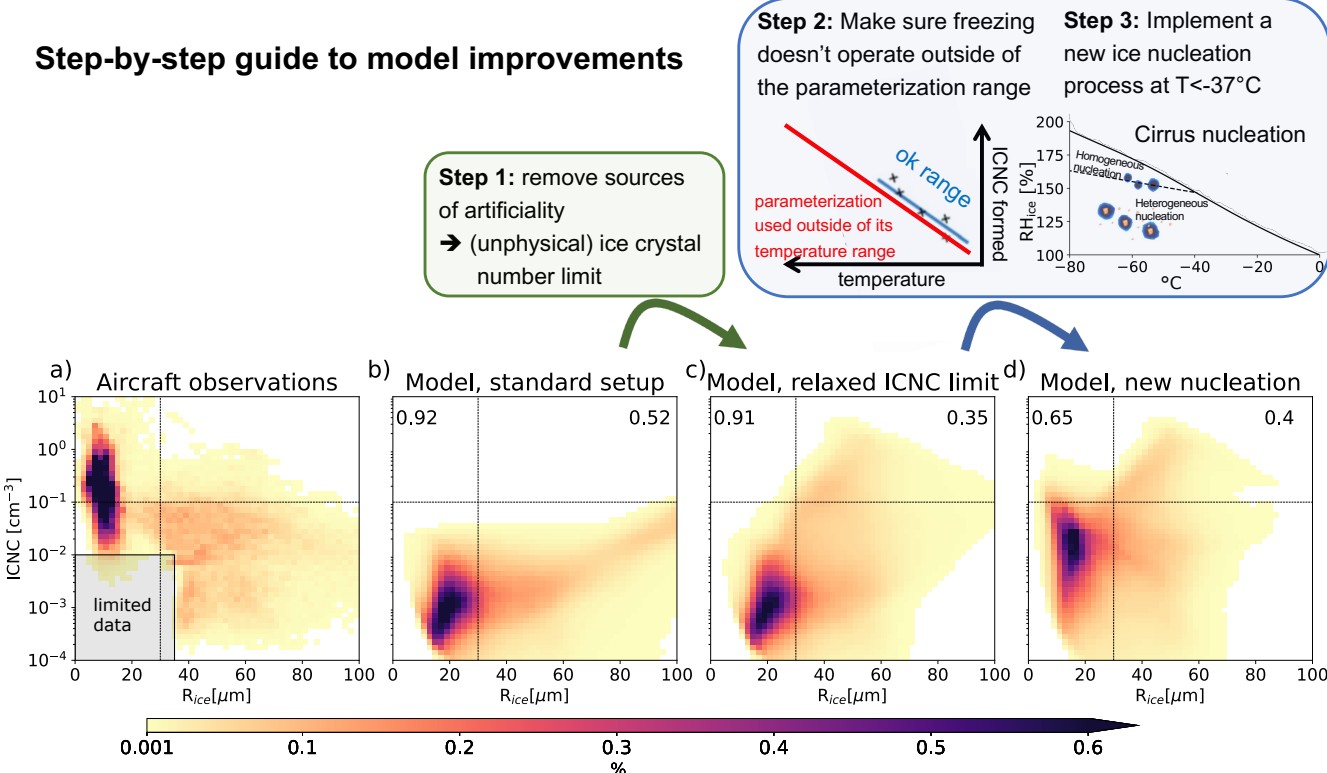

**Figure 5.** Probability density function of ice properties for clouds at T<-40°C for (a) tropical Pacific aircraft observations and three versions of the SAM model: (b) the standard setup, (c) the intermediate model version with a relaxed ICNC limit, and (d) the final version with a modified ice nucleation scheme. The number represents a 2D total variation distance of model data compared to aircraft observations (the smaller the number, the better the agreement), calculated separately for small and large particle sizes. Observations are limited or not available in the shaded area.

## 4 Cirrus properties

### 4.1 Simulated tropical cirrus cloud properties and their comparison with aircraft observations and satellite retrievals

This section first outlines the step-by-step changes implemented in the ice microphysics scheme in the ice number–radius space. We focus on changes under cirrus conditions, limiting the analysis to temperatures colder than -40°C and ice water contents larger than $10^{-5}$ g m$^{-3}$, which is close to the detectability threshold of the aircraft observations.

Aircraft observations in Fig. 5a show a peak ICNC between 1 and $10^{-2}$ cm$^{-3}$ and mean mass radii smaller than 30 $\mu$m. The mode extends to 3 cm$^{-3}$ at particle sizes smaller than 20 $\mu$m. Moreover, for concentrations smaller than $10^{-1}$ cm$^{-3}$, the

observed particle size often exceeds 50 $\mu$m. We note that due to retrieval limits, there are no measurements available for ice radii smaller than 35 $\mu$m at number concentrations below about $10^{-2}$ cm$^{-3}$ (see Methods).



The standard version of the SAM model coupled with the P3 scheme (Fig. 5b) is strongly biased compared to observations. Most notably, the model drastically underestimates ICNC as it lacks concentrations larger than $10^{-2}$ cm$^{-3}$. We resolve a large part of the bias by implementing three key changes to the ice microphysics.

We first relax the maximum ICNC limit from $5 \cdot 10^{-2}$ cm$^{-3}$ to 20 cm$^{-3}$ (Fig. 5c). This improves the representation of particles larger than 30 $\mu$m, reducing the total variation distance metric (Gibbs and Su, 2002, , a 2D analog to the root mean square error) from 0.52 to 0.35. However, the model still strongly underestimates the number densities of small ice crystals, indicating errors in parameterizing ice formation under cirrus conditions.

    The second modification addresses the deposition freezing parameterization, which was incorrectly active at temperatures

both below and above the homogeneous freezing threshold. Originally calibrated for temperatures warmer than -25°C (Fig. 1), this parameterization extended far beyond its intended range. We restrict it to T>-37°C and introduce a scheme to account for competition between homogeneous and heterogeneous nucleation at T<-37°C (Liu and Penner, 2005; Shi et al., 2015). This scheme captures low ICNC heterogeneous nucleation at RH$_{ice}$ > 120% while also allowing for homogeneous nucleation events under sufficiently strong updrafts and high RH$_{ice}$.

The two changes cut the microphysical bias in half, improving the representation of both small and large particles (Fig. 5d). Nevertheless, some substantial biases remain. The model continues to underestimate ICNC for small ice crystals and overestimate ICNC for larger crystals that represent freshly detrained particles (Fig. 4a-b). These remaining biases largely stem from persistent challenges in representing ice microphysics (see Discussion) and from too low vertical wind variance in the model (see Section 4.3).

To provide an alternative perspective, we examine the model's performance by sorting results by temperature. The exponential decrease in IWC with decreasing temperature, as expected from the Clausius-Clapeyron relationship, has improved compared to the standard model version and is well represented by the model (Fig. 6a-c). Observed ICNC shows large variability, with more than one order of magnitude increase in the median number between temperatures of -40°C and -70°C (Fig. 6d). The model reproduces the observed median and spread for temperatures warmer than -65°C but underestimates ICNC at the

coldest temperatures, although this bias has been substantially improved compared to the standard model version (only gray median shown). This bias aligns with the underrepresentation of small particles highlighted in Fig. 5. Insights from tracers (Fig. 4) suggest the bias may stem from too few in situ nucleated particles or insufficient detrainment of small particles from deep convection.

    Similarly to IWC, the mean mass radius decreases with temperature, with medians ranging from 80 $\mu$m at -40°C to 15 $\mu$m

at -80°C. The model simulates particles that are too small at warmer temperatures (between -55°C and -40°C) and slightly too large at T <-70°C. Notably, the spread in simulated particle size is narrower than observed, possibly due to the too simple single-mode description of ice microphysics (see Discussion).

    In summary, the temperature-sorted model results offer a complementary perspective on cirrus cloud properties, showing good agreement with observations for temperatures warmer than -60°C while highlighting persistent biases at colder temper-

atures. Nevertheless, this agreement represents a substantial improvement compared with the earlier model version (Fig. 5),







**Figure 6.** Cirrus cloud properties in tropical aircraft measurements (first column) and SAM model simulations with improved microphysics at 1 km horizontal grid spacing (second column). Lines represent median values. Gray lines in the middle column represent median values of the standard model setup. The third column shows the anomalies between SAM and aircraft data; the lines are copies of the lines on the first and second column panels. The data is sorted into 4°C temperature bins. The values in each temperature bin add up to 100%.





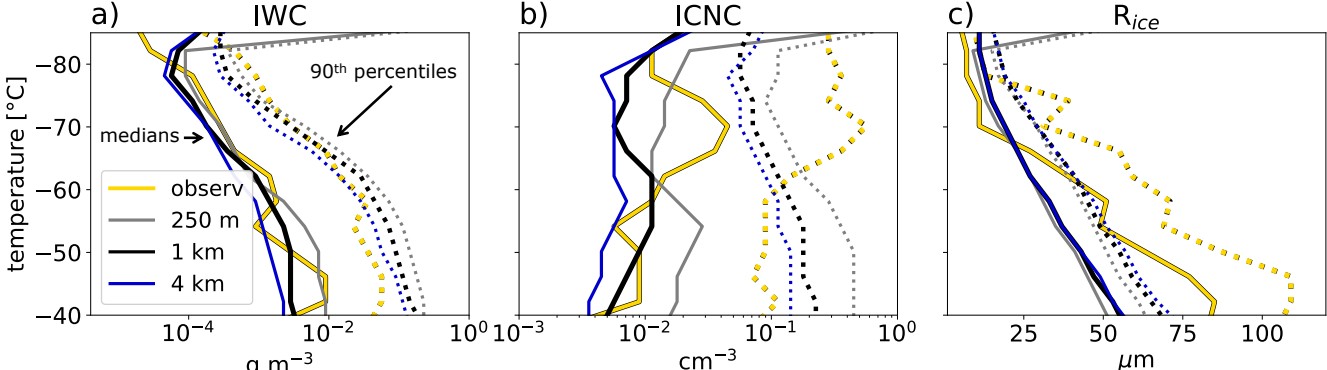

**Figure 7.** Grid spacing dependence of (a) ice water content (IWC), (b) ice crystal number concentration (ICNC), and (c) ice radius ($R_{ice}$) in SAM with improved microphysics compared to aircraft observations (in yellow). All data are regridded to a horizontal resolution of 4 km. Plotted are median values (solid lines) and 90th percentile values (dotted lines).

emphasizing processes where further refinement is still needed. The addition of passive tracers discussed in section 3 helps pinpoint processes requiring further refinement, particularly the representation of in situ ice nucleation at cold temperatures.

## 4.2 Horizontal grid spacing dependence

All results presented so far are based on simulations with a horizontal grid spacing of 1 km, which is sufficient to represent
anvil cloud evolution and their influence on mesoscale circulation (Gasparini et al., 2019, 2022). To assess the impact of model grid spacing, we include additional simulations at coarser (4 km) and finer (250 m) grid spacings. The 4 km grid spacing aligns with current GSRMs, while the 250 m grid spacing approaches the level of convergence for deep convective updraft strength in idealized tropical convection setups (Khairoutdinov et al., 2009; Jeevanjee et al., 2017).

The sensitivity of simulated ice properties to horizontal grid spacing is most pronounced for ICNC, which increases by
nearly an order of magnitude from the 4 km to the 250 m simulation, a trend also evident in the 90th percentiles (Fig. 7b). IWC exhibits grid spacing dependence only at temperatures warmer than -60°C, with higher values at finer grid spacings (Fig. 7a). In contrast, ice crystal radius shows little sensitivity to grid spacing (Fig. 7c). Moreover, SAM underestimates the variability in ice crystal size across all grid spacings and fails to reproduce the 90th percentile of observed particle size.

Overall, simulated microphysical properties at all grid spacings fall within the range of observations. However, the 4 km
simulations underestimate both IWC and ICNC across most temperatures, these biases are reduced in the 1 km simulation. The ice number–radius perspective provides a clearer view of these improvements, particularly for the low ICNC bias of ice crystals smaller than 30 $\mu$m (Fig. A1). Increasing grid spacing helps to better identify and understand the sources of model bias. The following section explores updraft variability, a likely contributor to these biases and a key factor in the reduced model bias observed for finer horizontal grid spacing.





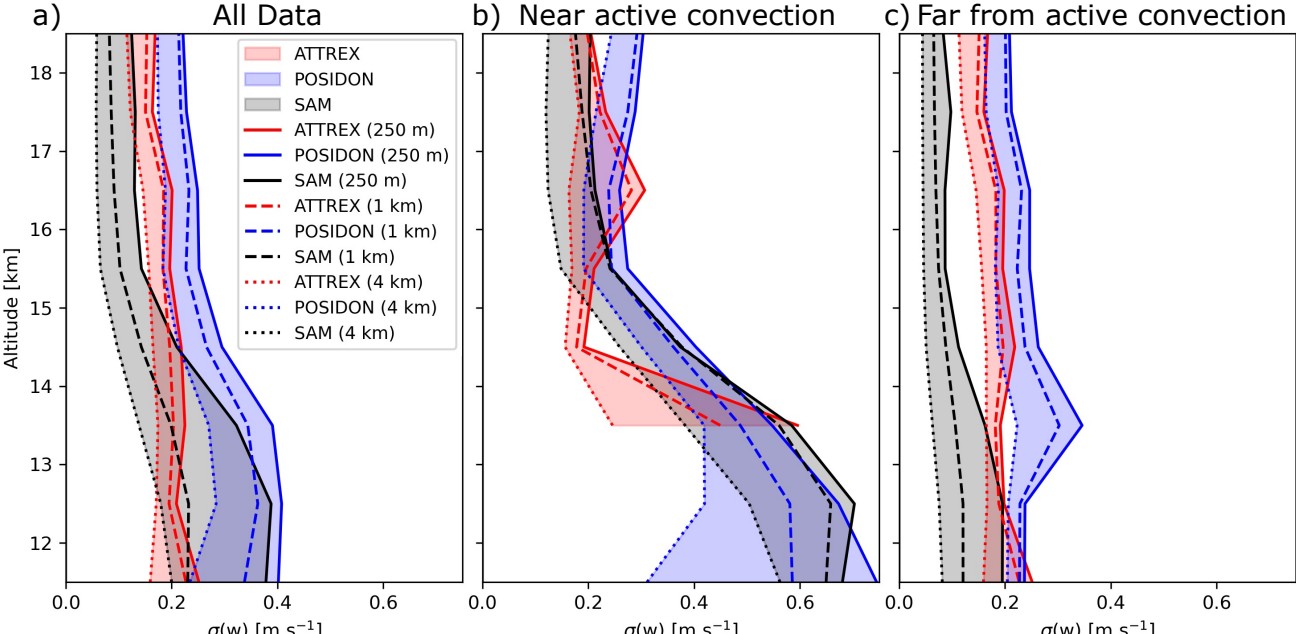

**Figure 8.** Standard deviation of updrafts from simulations with the improved SAM model and aircraft observations for (a) all data, (b) locations near active deep convection (brightness temperatures < 240 K), and (c) areas far from active deep convection (brightness temperatures > 240 K).

## 4.3 Vertical wind variability


Ice formation is strongly influenced by the availability of water vapor and on the supersaturation with respect to ice, which depends on the dynamical environment. Accurately capturing updraft variability is thus critical for modeling cirrus cloud formation and properties (Barahona et al., 2017). Of particular significance are the high-frequency fluctuations, which are approximately ten times larger than the slow synoptic scale motions (Atlas and Bretherton, 2023).

The biases in microphysical properties, especially ICNC, align with the model's underestimate of strong updrafts. This results in a too narrow updraft distribution and an underestimated standard deviation (Fig. 8). Including a subgrid-scale updraft velocity term minimally improves the issue slightly by increasing the standard deviation in updrafts by 1-10% depending on the grid spacing (2-4% for 1 km grid spacing). Nevertheless, the model performs well in representing the updraft variability near regions of deep convection at all three horizontal grid spacings (4 km, 1 km, 250 m) (Fig. 8b).

However, updraft variability is strongly underestimated in regions far from deep convection (defined as areas with brightness temperature > 240 K) for simulations at 4 and 1 km grid spacing. The 250 m simulation shows better agreement with the measured winds below 14 km, where anvil cloud coverage is the largest, but underestimates wind variability in the TTL, resultign in a persistent ICNC bias for small ice crystals (Fig A1c and f). This also likely contributes to larger microphysical biases in clouds far from active convection compared to those near convection (Fig. A2).



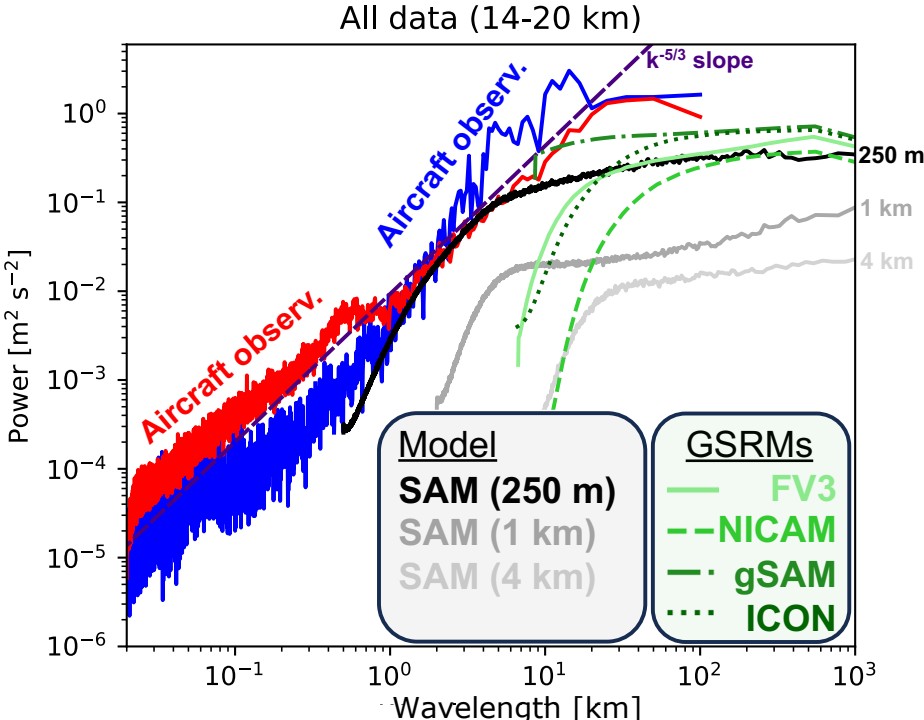

**Figure 9.** Power spectrum of updrafts from ATTREX (red) and POSIDON (blue) aircraft measurements, improved SAM model simulations (in grayscale), and a selection of GSRMs (in green). The dashed pink line represents the $k^{-5/3}$ slope, where k is the wavenumber.

Power spectra in Fig. 9, calculated following Atlas and Bretherton (2023), provide further insight into updraft variability accross different wavelengths of atmospheric disturbances. Variability at wavelengths larger than ∼1000 km corresponds to synoptic-scale motion, while convectively generated gravity waves dominate at wavelengths between 1 and 1000 km. Turbulent processes of various sources dominate at wavelengths below ∼200 m.

    As already shown by Fig. 8, wind variance increases with finer horizontal grid spacing. While the 4 and 1 km simulations

substantially underestimate wind variance and power, the 250 m simulation reproduces the observed wind variability for wavelengths between 1 and 10 km. Moreover, all SAM simulations under-represent variability at scales larger than ∼100 km, which are better captured by GSRMs at horizontal grid spacings of about 4 km. Gravity waves, originating from deep convective updrafts, propagate hundreds to thousands of kilometers from their source. However, the narrow channel setup of the SAM simulation restricts the generation of these waves compared to the real atmosphere or global GSRM simulations. While larger-

scale updraft variability fosters favorable conditions for deep convection, its direct contribution to cloud formation is limited and of secondary importance for this study.





**Figure 10.** (a) Frequency of cloud occurrence in each ice water path (IWP) bin, (b) average CRE in each IWP bin, and (c) the contribution of each IWP bin to the CRE. The gray shading in panel a) highlights the range of observations. Panels b) and c) show results only for the improved SAM model and the CCCM satellite product. Model results are presented in its full resolution, and regridded to 36×36 km grid, similar to the resolution of CERES satellite retrievals of CRE.



## 4.4 The relevance of tropical cirrus for top-of-the-atmosphere radiative fluxes

To link cirrus cloud properties to the TOA radiative budget, we take an integrated perspective and categorize cloud occurrence frequency and radiative effects based on IWP. Figure 10a compares the simulated IWP distribution with four satellite-derived datasets, showing that the improved model generally falls within or near the observed range. The apparent overrepresentation of the thinnest cirrus in the model and the corresponding slight underestimation of other cloud types in the normalized PDF should not be interpreted as a bias, as satellite retrievals are known to underestimate the thinnest cirrus clouds (Balmes and Fu, 2018; Lesigne et al., 2024). We also find that thin cirrus with IWP $< 1$ g m$^{-2}$ constitute 52% of all simulated cirrus but contribute only 6% to the LW CRE and 5% to the SW CRE. This suggests that while thin cirrus are frequent, their direct radiative impact at the TOA is limited. Restricting the analysis to the most radiatively important clouds with IWP $> 1$ g m$^{-2}$ improves the model agreement with satellite observations (Fig. B1).

Figure 10b shows the averaged CRE for high clouds in each IWP bin. The most frequent anvils according to satellite datasets occur at an IWP of 3-30 g m$^{-2}$, which corresponds to anvil clouds of intermediate optical depth (1-3) that yield a net positive, LW-dominated CRE (Sokol and Hartmann, 2020). For thicker clouds (IWP $> 100$ g m$^{-2}$), both model and satellite data show a dominance of SW CRE, with net CRE exceeding -100 W m$^{-2}$ for the thickest anvils. However, the model overestimates the SW CRE, a bias partially resolved by averaging the output onto coarser grid scales, similar to the CERES pixel data resolution (36×36 km grid boxes). Additionally, a recently identified bug in the ice optics parameterization likely increased the optical depth and radiative effects per unit IWP by ∼15%, but this issue could not be addressed in the current simulations. This SW bias becomes even more apparent when comparing CERES CRE retrieved during daytime satellite overpasses with model-simulated CRE between 1 and 2 pm local time (Fig. B2b).

Determining which type of tropical cirrus is radiatively most important is not straightforward. Are the less frequent but thick deep convective cores and fresh anvils, which have a strong influence on both SW and LW CRE, the most dominant, or do the more widespread anvil clouds of intermediate thickness dominate? Figure 10c provides an answer by scaling the CRE in a given IWP bin by that bin's frequency of occurrence: the radiatively most dominant clouds have IWP between 30 and 3000 g m$^{-2}$, similar to results by Berry and Mace (2014). These include anvil clouds with optical depths greater than approximately 1. Nonetheless, thinner cirrus (IWP 1–30 g m$^{-2}$, Fig. 3c) also contribute meaningfully to the net CRE, underscoring the importance of studying not only the thickest tropical cirrus but also their continued evolution until they reach an IWP of ∼0.1 g m$^{-2}$ (optical depth ∼0.1). Additionally, the response of thin cirrus to global warming remains highly uncertain (Sokol et al., 2024), and deserves further investigation.

This analysis, together with our previous findings (Fig. 3), provides more information about the evolution of tropical cirrus. These can be split into two separate evolution pathways, as depicted in Fig. 10a. Thick anvils originating in high IWP deep convective towers spread and thin towards IWP ∼1 g m$^{-2}$, while thinner cirrus (IWP $< 1$ g m$^{-2}$) predominantly form via in situ ice nucleation. Despite their smaller radiative impact, these thin cirrus are highly frequent and likely play a much larger role in shaping TTL temperatures and influencing deep convective overshoot frequencies (Fu et al., 2018; Hu et al., 2021). Both phases are associated with substantial uncertainties in their microphysical properties and radiative impacts. Addressing



these uncertainties is essential for improving our understanding of cirrus feedbacks in the climate system and will be the focus of follow-up studies.

## 5 Discussion

### 5.1 Ice microphysics

The representation of ice microphysics remains a limitation in this study, despite notable improvements achieved with the described computationally inexpensive changes. This reflects the broader challenge of accurately modeling ice microphysics in high-resolution models, where computational constraints and incomplete process understanding often impose trade-offs.

For example, the adopted Shi et al. (2015) scheme for ice nucleation, originally designed for coarse climate models, relies on empirical fits to parcel model simulations. While effective, this approach may not capture all the details of ice nucleation

at cloud-resolving scales accurately. Future work could address this by implementing a physics-based nucleation scheme, such as the novel Kärcher (2022) scheme. Moreover, a bug was discovered in processing of in situ observations after we had completed the simulations and this prevented us to resolve the model's issue of underestimating small particles with high ice crystal number concentrations. While model tuning could partially resolve this issue, a more robust solution would involve implementing a more realistic nucleation scheme and/or incorporating a resolution-aware parameterization of wind variability

to complement the model-resolved wind variability – similar to approaches being developed for liquid clouds (Salesky et al., 2024).

Secondary ice formation may be another area of interest, given that recent studies have shown that it plays a large role even at temperatures as cold as -50°C (Hawker et al., 2021; Huang et al., 2022; Qu et al., 2022). Incorporating secondary ice processes into SAM-P3 currently requires additional computationally expensive ice modes, but their inclusion could enhance

the model's ability to simulate anvil cloud evolution. Finally, increased model complexity brings additional challenges, often resulting in hindered process understanding, increased uncertainty, or the problem of equifinality (refer to Proske et al. (2023) and references therein for a complete outline of these issues).

An alternative approach could involve introducing ice tracers for each ice nucleation process, similar to those used by Lüttmer et al. (2024). This would allow for a more accurate classification of cirrus cloud origins without relying on arbitrary

thresholds. Although this would modestly increase the model's computational costs, the expense is substantially lower than doubling the model's resolution, making it a viable path forward.

### 5.2 Resolution and Lagrangian approaches

Our findings confirm that increasing horizontal resolution improves the representation of atmospheric updrafts at scales crucial for cirrus cloud formation. However, this improvement comes at a large cost: a 4-fold increase in horizontal resolution results

in a 16-fold increase in data output and a 25-30-fold increase in computational demand. This highlights the importance of exploring less computationally expensive alternatives to improve model performance. For instance, we achieved a large reduction





in bias for ice cloud properties through straightforward modifications to the ice microphysics without appreciable increases in computational cost.

Encouragingly, the simulations at a typical GSRM grid spacing of 3-5 km show signs of convergence in updraft variability near active deep convection when compared to updraft observations averaged to same grid spacing, suggesting that the generation of convective gravity waves is reasonably well captured (note that model-simulated updraft variability still increases between simulations of 4 and 1 km grid spacing). However, these waves do not propagate far enough, as indicated by the underestimated updraft variance across all tested horizontal resolutions. Power spectral analyses indicate that grid spacings slightly finer than 250 m may sufficiently capture the scales of motion relevant for cirrus formation and maintenance. This is also the grid spacing at which convergence in cloud macroscopic variables and ice sources and sinks has been observed (Hu et al., 2024).

Vertical resolution also plays a crucial role. Previous studies demonstrate its influence on tropical cirrus properties, their responses to global warming (Ohno et al., 2019), and convective self-aggregation (Jenney et al., 2023). We also perform a short sensitivity test that doubles the vertical grid spacing from 200 m to 100 m increased updraft variability away from deep convection while raising computational costs by only a factor of 4.5 (not shown). This suggests that refining vertical resolution could be a more computationally efficient way to improve model updraft variability than increases in horizontal resolution, and it should be thoroughly investigated in future studies.

Nevertheless, higher resolution alone cannot address all challenges. Processes at microscopic scales, particularly the interactions between ice microphysics and radiation, remain poorly resolved. Efforts to improve ice microphysics in GSRMs are scarce (e.g., Seiki and Ohno, 2022), yet essential for advancing understanding of tropical cirrus evolution. Our study highlights the importance of accurately modeling cirrus evolution, a key factor in determining their microphysical properties and radiative effects.

Passive tracers, as demonstrated in this study, are a valuable tool for tracking the evolution of ice clouds in models. Although these tracers are purely computational, stable water isotopes, measurable in situ or via satellite, could serve as real-world tracer analogs, offering insights into the pathways of ice cloud evolution (Blossey et al., 2010; de Vries et al., 2022). Trajectory analysis, such as that in Sullivan et al. (2022), can provide additional clarity on cirrus cloud evolution. Based on their analysis of cloud source and sink processes along trajectories, they proposed three cirrus cloud lifecycles with distinct radiative signatures. In contrast, our analysis focused only on ice cloud properties. However, by enabling the 3D output of microphysical process rates, we could easily perform a similar process-rate analysis that would provide additional clarity on the two tropical cirrus formation pathways. Future work could explore a consistent integration of these approaches in models and observations, linking simulations with aircraft measurements as in Froyd et al. (2022) and possibly satellite retrievals as well, in an effort to improve process understanding.



## 6 Conclusions

The purpose of this work is twofold: first, we demonstrate the usefulness of passive tracers to track the evolution of cirrus
microphysical properties. Second, we show that minor changes to cloud microphysics that substantially improve the simulation
of tropical cirrus at cloud-resolving scales with a minimal change in computational expense.

Our work reveals a simple, numerically inexpensive recipe that substantially improves simulations of tropical cirrus in the
SAM cloud-resolving model. Although the recipe was tested only in this specific model, we believe that its elements can be
applied to a wider range of models with little or no increase in computational load. The recipe is as follows:

1. Remove all non-physical microphysical limits (particularly the commonly used ICNC limits, see e.g. Bacer et al., 2021).

2. Ensure that freezing (or any other) parameterizations do not operate outside of their stated range of validity (e.g. limit
   mixed-phase freezing parameterizations to mixed-phase conditions).

3. If not present, add a nucleation scheme that is active under cirrus conditions, e.g. homogeneous nucleation of water
   solution droplets or a combination of homogeneous and heterogeneous nucleation.

We show that anvil clouds remain radiatively important hours after detrainment, after having undergone substantial spread-
ing, thinning, and advection by synoptic and/or mesoscale motion. Therefore, an evolutionary perspective on tropical cirrus is
crucial for constraining their radiative impacts. The implementation of passive tracers enables an evolutionary perspective on
tropical cirrus clouds. We implemented two passive tracers to track the three-dimensional evolution of cloud parcels through
two distinct perspectives, namely:

1. A detrainment perspective, useful for tracking the evolution of anvil clouds.

2. An ice nucleation perspective, useful for tracking the evolution of in situ cirrus.

Tracers also provide important insights into the climatology of cirrus cloud formation. We find that in situ cirrus dominate
under colder conditions (70% of cirrus at temperatures below -70°C) and are prevalent at low ice water path values (IWP<
1 g m$^{-2}$). Despite their low optical depth, these clouds cannot be neglected in the TOA radiative balance analysis. However,
our estimates are sensitive to the specific classification criteria used for cloud origin, highlighting the need for more refined
approaches. A more accurate, but also computationally more expensive model setup with multiple ice species will address this
uncertainty in a follow-up study.

Our results also suggest that simulations with horizontal grid spacing of 250 m can reproduce the observed power spectrum
of vertical wind, capturing the scales of motion relevant for cirrus formation and maintenance, particularly below 14 km.

More broadly, our work highlights a significant limitation in existing global climate and cloud-resolving models, many of
which fail to accurately simulate tropical cirrus evolution (Wall and Hartmann, 2018; Turbeville et al., 2022; Atlas et al., 2024).
These biases prevent the understanding of the processes that occur between the formation of tropical cirrus and their dissipation
(Gasparini et al., 2023), leaving large uncertainties in the associated climate feedbacks (Sherwood et al., 2020). Additionally,



the interaction of ice crystals with radiation, which influences atmospheric temperatures, can alter cloud lifetimes (Gasparini
et al., 2022), regional climate (Voigt et al., 2019), and hydrological sensitivity (McGraw et al., 2025), with potentially important
impacts as high clouds shift upward in a warming climate (Voigt et al., 2024; Gasparini et al., 2024). By addressing these gaps,
we have demonstrated that our improved SAM model is now equipped to explore these feedbacks with greater confidence.

Building on the advancements presented here, future studies should focus on reducing uncertainties in the fundamental
understanding and modeling of cirrus properties and their evolution in the present and in a warmer climate. The combination
of passive tracers, improved microphysics, and high-resolution modeling provides a promising pathway to achieving this goal.



## Appendix A: Additional model evaluation

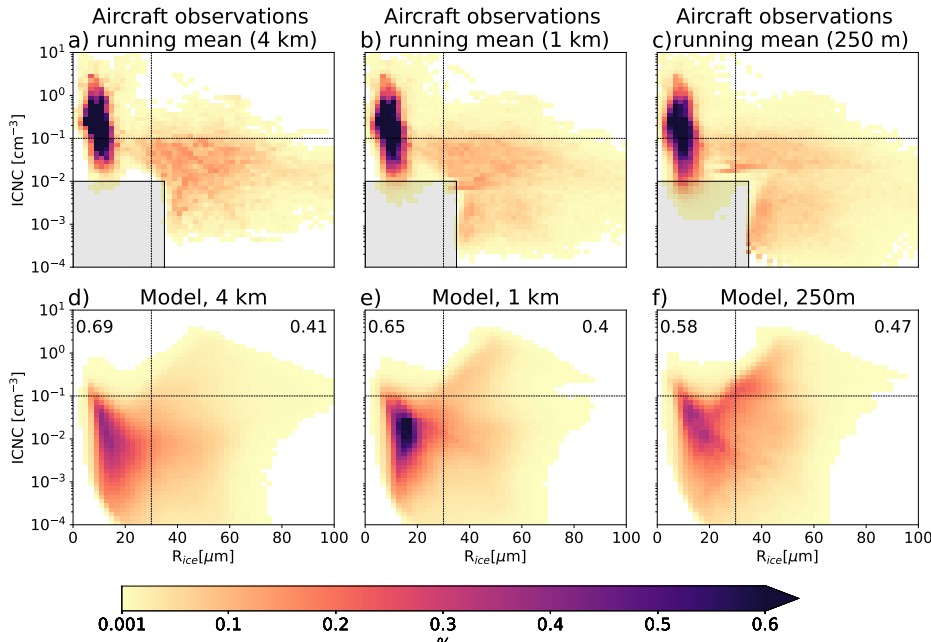

**Figure A1.** Model grid spacing dependence in the ICNC-$R_{ice}$ space. Probability density function of ice properties for clouds at T<-40°C. The upper row shows aircraft observations averaged for consistency to 4 km, 1 km, and 250 m grid. Panels d-f show modeled properties at different horizontal resolutions. The number represents a two-dimensional total variation distance of model data compared to aircraft observations (the smaller, the better). Observations are limited or not available in the shaded area.



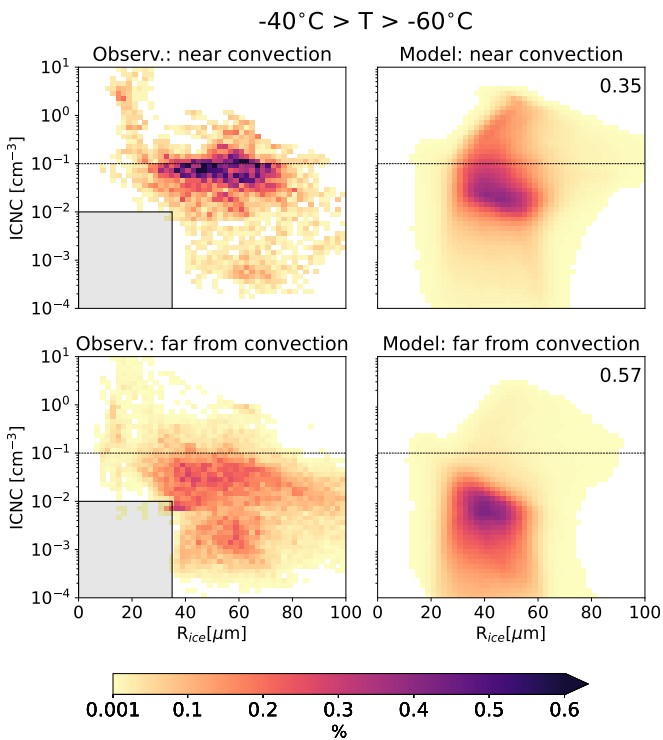

**Figure A2.** Probability density function of ice properties near (brightness temperature < 240 K) and far from active convection (brightness temperature > 240 K) at -40°C> T >-60°C. The number on the right is the dimensional total variation distance of model data compared to aircraft observations (the smaller, the better). Observations are limited or not available in the shaded area.

## Appendix B: Additional IWP-binned perspective on model output and satellite observations

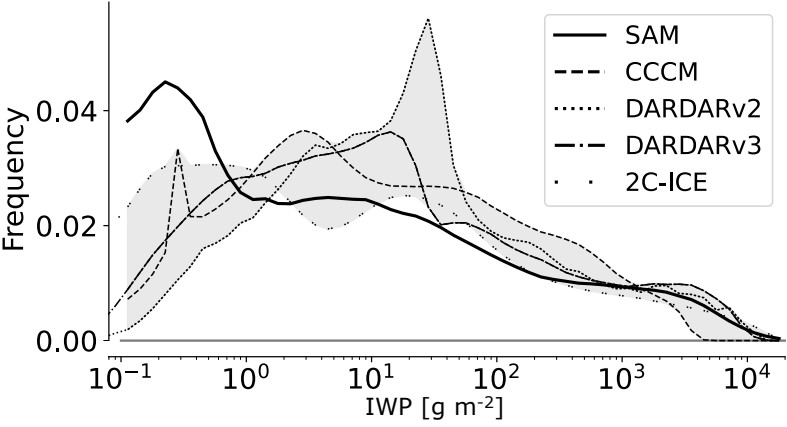

**Figure B1.** Ice water path (IWP) binned occurrence frequency. As in Fig. 10a, but for IWP>0.1 g m$^{-2}$.




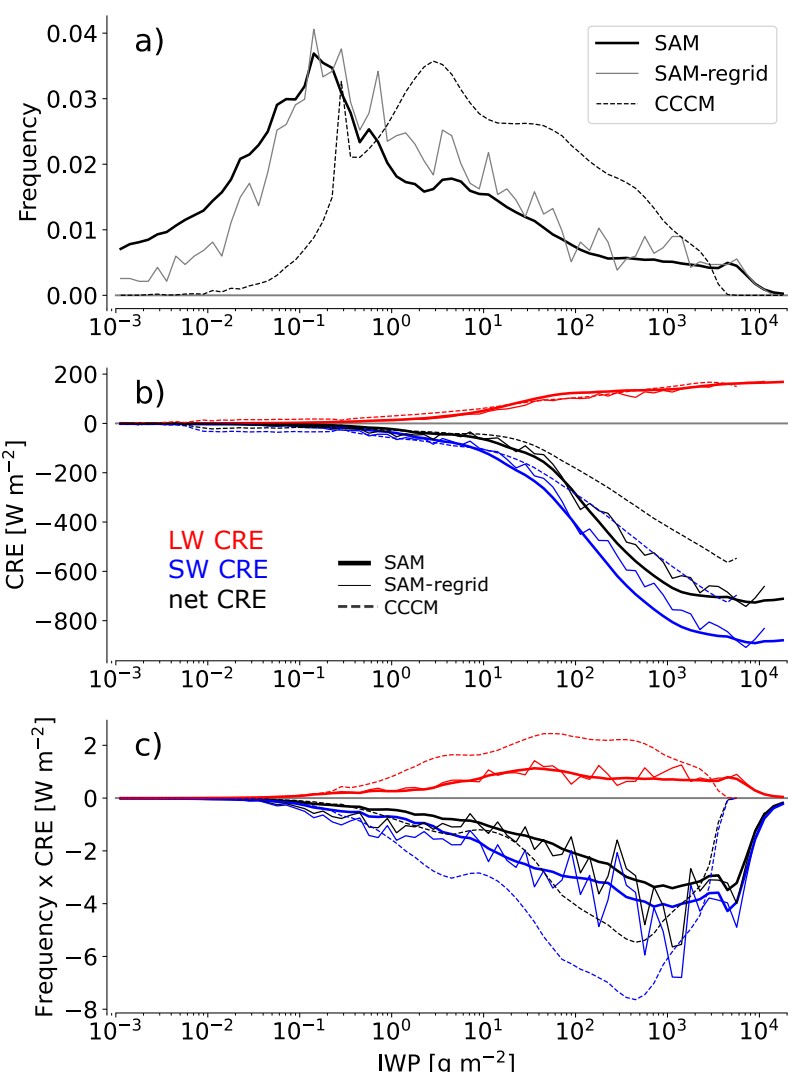

**Figure B2.** Daytime-only ice water path (IWP) binned cloud occurrence and radiative effect. The figure is similar to Fig. 10, but with (1) model results for 1 and 2 pm local time only and (2) CCCM SW radiative fluxes computed as a difference between clear and full sky radiation.



*Code and data availability.* Data, plotting and post-processing scripts, key (modified) bits of the SAM model Fortran code are available on Zenodo at https://doi.org/10.5281/zenodo.14674413. THe complete SAM model code is available for download at https://you.stonybrook.
edu/somas/sam/. Satellite data from the A-Train Integrated CALIPSO, CloudSat, CERES, and MODIS Merged Product Release B1 (CCCM) were obtained from https://search.earthdata.nasa.gov. The DARDAR data are available at http://www.icare.univ-lille1.fr/. The 2C-ICE data are available at https://www.cloudsat.cira.colostate.edu

*Author contributions.* B.G.: conceived the study, implemented code changes, conducted model simulations, processed the majority of the data, and contributed to writing and editing; R.A.: processed part of the data and contributed to writing and editing; A.V.: contributed to
writing and editing; M.K.: acquired and processed data, and contributed to writing and editing; P.N.B.: implemented code changes, provided technical support, and contributed to writing and editing.

*Competing interests.* One of the co-authors is a member of the editorial board of *Atmospheric Chemistry and Physics*. The authors have no other competing interests to declare.

*Acknowledgements.* BG and AV acknowledge funding received from the European Union's Horizon 2020 research and innovation pro-
gramme under the Marie Skłodowska-Curie grant agreement No 101025473. This material is based upon work supported by the National Science Foundation under Grant No. OISE-1743753 (PB). We would also like to acknowledge high-performance computing support from Cheyenne (https://doi.org/10.5065/D6RX99HX) and Derecho (https://doi.org/10.5065/qx9a-pg09) provided by NCAR's Computational and Information Systems Laboratory, sponsored by the National Science Foundation. The authors would like to thank Bernd Kärcher for useful comments on early work leading to this manuscript. We acknowledge the use of several AI tools to improve the language and clarity of this
manuscript.



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
