# Peer review of "Tropical cirrus evolution in a km-scale model with improved ice microphysics"

_EGUsphere, 2025_

## Author Comment (AC2)

**Response to reviewer's comments, Gasparini et al., 2025**

We thank the reviewers for their valuable comments.

First, we would like to highlight larger changes in the new version of the manuscript, mainly (but not only) driven by the comments.

The most important changes are listed here, other comments are listed directly in the response:

1. The observational data for ATTREX and POSIDON has changed. The original manuscript used a version of the data that contained a bug which led to falsely large ice crystal numbers and small radii. After fixing the bug, the data are almost identical to the version we used for tuning purposes prior to the initial manuscript submission. As expected, the model performs much better with the current data than with the data from the initial submission. That impacted figures 2,3, 8, A1, A2 in the new manuscript version.

2. The cloud classification mechanism has been adapted to account for the effect of ice sedimentation, which leads to misclassification of dual-origin cirrus as in situ cirrus. Tracers follow air parcels, while ice crystals sediment out of these parcels. Occasionally, this leads to an unphysical, odd classification of cloud categories, especially affecting the longer-lived clouds. In such cases, the in situ cirrus tag has often been assigned to the lower parts of clouds. Such clouds generally have the same microphysical origin; the classification scheme splitting clouds, e.g. into dual-origin and in situ (see panel d, Figure below), was therefore an artefact of the intrinsic limitations of a pure tracer-based cloud classification. To correct this behavior, we reclassified all cloud grid boxes that were classified as in situ cirrus and were below dual-origin cirrus grid boxes as dual-origin cirrus (see panel e in the figure below).

This still leaves a problem of occasional misclassification of some scattered cloudy gridboxes as anvils; however, we postpone additional fixes to the cloud classification algorithm to future work.

In addition, the threshold time for a cloud to be classified as in situ is reduced from "time after convection" of 30 hours to 24 hours.

While the described changes are impactful on the example shown, the key outcomes of the statistical evaluation of cloud origin remain robust (confront former figure 3 in the submitted version of the manuscript with the figure 6 in the revised manuscript).

The description of cloud categories was also adapted accordingly in text.

*To better characterize the microphysical origin of cirrus, we classify them into three categories: in situ, anvil, and dual-origin. Anvil cirrus are defined as clouds where the time since detrainment is shorter than the time since in situ nucleation. In situ cirrus are those that have not experienced detrainment for at least 24 hours and where the time since nucleation is shorter than the time since detrainment.*

*Dual-origin cirrus are clouds detrained within the last 24 hours that have a shorter time since in situ nucleation compared to time after detrainment. These typically form within or near anvils or in cirrus clouds close to active convection (Fig. 5d). To correct for classification biases due to ice crystal sedimentation impacting older anvils and in situ*

*cirrus, clouds that were initially classified as in situ cirrus but located below dual-origin cirrus are reassigned to the dual-origin category. Although dual-origin cirrus are influenced by in situ nucleation, their total ice mass and number are still dominated by convective outflow (not shown).*

Additionally, we created an animation demonstrating how the classification performs beyond the timestep depicted in Figure 5. It extends over five simulation days and can be found in the supplementary data.

3. At the suggestion of the review team (reviewer 1) we decided to reverse the order and present section 4.1 first, followed by section 3 (tracer analysis) and then sections 4.2 and 4.3. Because of this, the "diff" file is often misinterprets changes to the manuscript.

[Figure]

REVIEWER 1

This review was produced by a co-review team, as agreed with the editor.

General comments

The authors study tropical cirrus using an idealised simulation with a convection-permitting model (SAM), multi-campaign aircraft dataset (POSIDEN and ATTREX), satellite data (DARDAR, CCCM and 2C-ICE), and a selection of global-storm resolving models (GSRMs). It analyses these data from a variety of angles, including the use of atmospheric tracers, power spectra, standard deviation of vertical velocity, cloud radiative effect, ice water content (IWC), ice particle concentration and ice particle radius. The tracer analysis in particular focuses on the distinction of cirrus detrained from convective anvil cloud opposed to cirrus formed in-situ. Following this analysis, the authors conclude that in-situ cirrus prevails at colder temperatures and low IWC, while anvil cirrus prevails at warmer temperatures above -60C. The radiative analysis highlights the anvil cirrus as particularly important, but the authors suggest in-situ cirrus is still important in this regard. The authors explore model resolution and show that the updraught power spectra (which they consider important for cirrus formation) only becomes accurate around 250m grid spacing. They achieve better agreement between the model and observations with their proposed methods without the need to increase the computational cost.

It is great to see multi-campaign aircraft data exploited for improving model performance, and it is great to see some microphysical model development with computational resource in mind. There are many interesting results in this paper and, while they jump around different aspects of cirrus research, we think there is enough coherence for it to make a paper. However, if trying to cover a lot of ground, a paper needs a structure that helps the reader. We don't think that is quite the case here. The least subjective aspect of this is the presentation of model improvement content after the presentation of scientific model results in relation to the tracers (which must have been based on the model improvements). We cannot understand the logic in this, so we suggest moving section 3 after at least section 4.1 but better would be after the whole of section 4. Other than this, we take issue with a few statements by the authors and believe there are few possible improvements, but once that has been considered we hope to see the manuscript reach publication.

The reason for having Section 3 before Section 4 is simply to highlight what we think is the main novelty of this paper. We agree that it makes more sense to evaluate the model before describing the tracer, so we have reversed the order.

**Major comments**

**L9/L361** – The result suggests that in-situ, thinner clouds are not radiatively important. However, the authors caveat in both the abstract and in the text that such cloud must still be considered when looking at such radiation. The manuscript does not show any results to back this up, and I cannot see any argument made for why that is the case. Why "must" the in-situ CRE be considered? In what way does the thinner cirrus "contribute meaningfully"?

This is an important point, and we clarified it in the revised manuscript. We multiplied the PDFs for each cloud class separately (see Fig. 6c) by the averaged CRE in each IWP bin shown in Fig. 4b. We added the values in W/m² and the relative contribution of each cloud class to the total CRE in a separate table described in Section 4.2.

While our simple analysis provides approximate CRE values, a proper attribution of radiative effects should consider (1) cloud overlap (see the relevant discussion in Deutloff et al., 2025 [doi: 10.1029/2024MS004615], regarding anvil radiative effects) and (2) the partial classification of cloud portions as anvil origin, as shown in panel e) of this reply document.

We plan to improve the cloud classification method in future work. This enhancement will use morphological post-processing methods, e.g., adapting one of the existing 3D cloud tracking algorithms.

The radiative contribution of clouds at IWP $< 1$ g m$^{-2}$ to the total CRE budget is about 6% for SW CRE and about 7% for LW CRE. This part of the IWP distribution is dominated by in situ cirrus, which contribute to a roughly similar radiative contribution. See also the relevant discussion in the new Section 4.2 for more information.

**L270** – "cut the microphysical bias in *half* " – I cannot see what this is based upon. In Figure 5, the small particle bias metric goes from 0.92 to 0.65, and the large particle bias metric goes from 0.52 to 0.4. How is this halving? Perhaps you need to give the formula for the metric or explain it more. We don't fully understand what it is.

This was a left-over from the text and respective figures before the introduction of the erroneous bug fix of the observational data that happened just before the first submission (see general comment 1 in this reply document). The observational data returned to a version nearly identical to the one before the "bug fix", and thus the statement becomes valid again and is kept in text.

**L276** – CC applies to the saturation vapour pressure. Whilst I can understand that it plays a role in moisture with height in a stable atmosphere, the connection to anvil cirrus seems much more complicated, as the available moisture at each height is a result of CAPE and low-level humidity. In fact, I might naively assume that because the saturation vapour pressure reduces with height (i.e. temperature), one should expect increasing condensate as moisture is lifted as it cannot remain as vapour. Are you drawing this statement of CC yourself? In which case, please explain the logical steps in greater detail. If this is something you consider to be widely accepted, please point to a reference or two that explains the mechanistic connection between IWC vertical profile and CC.

We mentioned the statement by following results of the van Diedenhoven et al 2020 (doi: 10.1029/2019JD031811). Based on their work, the Clausius-Clapeyron control of ice cloud properties should hold for the most frequent anvil clouds (COD<20); however, as they focus on ice crystal size, and not ice water content, we decided to remove that statement from the paper. Moreover, we found observational evidence of fresh anvils behaving quite differently from what expected by Clausius-Clapeyron (Schiller et al., 2008, doi: 10.1029/2008JD010342).

**Fig9** – The GSRM data has not been described. Please add a description to the data section.

Included in text, lines 199-206.

**Fig 10** and **L366** – I strongly object to the illustration on fig10 and the suggestion of any analysis "split into two separate evolution pathways". Until this point, the paper was doing a brilliant job of both simplifying the problem whilst maintaining necessary complexity (e.g. Figure 3c).

The "dual-origin" category is somewhat arbitrary because it includes aged anvils that have been affected by ice nucleation. Therefore, it is reasonable to consider only two distinct evolutionary pathways that dominate different parts of the IWP space. Indeed, there is some overlap between the categories. Thus, we have extended the anvil sketch to include the anvil cirrus up to an IWP of approximately 0.1 g/m² (see Figure 4 in the text). Also see the relative perspective on the sources in the lower figure. In our opinion, the relative proportions of in situ versus all other cirrus are clear enough to maintain the sketch and related discussion.

[Figure]

However, as far as I can tell, you have not used the tracers in the analysis of CRE. This is unfortunate, and I would strongly encourage to break the CRE analysis down by tracer if possible, as it would be fascinating. However, you have instead subjectively drawn on clouds to Figure 10. These are misleading because there is not such a simple split. Your own Figure 3c shows the large amount of overlap in the central part of the IWP spectrum. Whilst you could adjust the illustration to show that in-situ and anvil cirrus occur in the same place, how will you illustrate the dual-origin component? I often appreciate diagrams, but I think it is necessary to remove this as I don't think it adequately reflects the value of your results.

We added a breakdown analysis for CRE into tracers and a short discussion (see the new Section 4.2).

**L367** – Fig 3c shows similar amounts of anvil cirrus all the way to 0.01 g m-2. This is why I think your illustration is misleading (as described above). It makes one think there is this clear distinction, but there is not. Just because in-situ cirrus dominates <1gm-2 cloud doesn't mean that a major fraction of anvil cirrus is not spreading to that thinness.

We interpret a 70:30 split shown in figure above as a clear origin distinction.

Minor comments

**L49** – In the outline of the paper you first say you demonstrate modification and then look at tracers. I suggest this is a more logical ordering for your sections than you actually have.

Done.

**Sec 2.1.2** – Possibly it would be good to further subsection this so each modification is clearly labelled as it would make it easier to refer back to the relevant text in fig5 caption.

Done.

**L167**- They use nighttime data only because the lidar signal is noisier during the daytime. Is this representative for the simulation?

This leads to some bias in the data. However, given that we are examining a region that is fully covered by the ocean, the diurnal cycle of high clouds is quite small, so the bias due to the diurnal cycle will be smaller compared to the retrieval uncertainty, which relies on many assumptions. Based on past modeling studies, we expect the cloud properties of anvils to not change drastically due to the diurnal cycle, although their longevity may change (e.g., Wall et al., 2020, doi:10.1175/JCLI-D-20-0204.1). Wall et al. (2020, observational perspective) and Gasparini et al. (2022, modeling perspective, doi: 10.1175/JCLI-D-21-0211.1) provide a longer discussion on the diurnal cycle of tropical oceanic anvils.

**Fig1** – Add description of solid and dashed black lines.

Done.

**Sec 2.2** – How long is the simulation and how much of that has been used in the results?

Added.

**Sec 2.3** and **2.4** – Maybe sections 2.3 and 2.4 would be better grouped under a "Data" section. The title for section 2.3 could be shortened to "In-situ observations". In section 2.4, there's no need for subsections, it's clear enough if each of the datasets is presented in separate paragraphs.

Model data is also part of the data. We changed Section 2.1 to better clarify changes in the model code (Sections 2.1.1-2.1.5), as suggested by another reviewer.

**Fig 2** – Could a second, later time be shown once convection has subsided? It would be useful to see how such cloud and tracer categorisation progresses after convection finishes but high cloud remains. It would be fine if only panel **d** was shown for the later time.

Good idea! We created an animation depicting the evolution over the course of five days. Note that the 3D model output was saved every six hours, and one output snapshot had corrupted tracer data. This animation is now part of the supplementary material. Examining

more instantaneous snapshots also motivated us to slightly change the classification criterion.

**L219** – "excludes portions" – what portion is excluded? I don't recall that being described.

This was just our interpretation of the results, not related to the classification criterion. It was confusing, and therefore we removed it from the text.

**Fig 3** – What's the length of the time period used to generate this figure?

20 days, now mentioned in the description of simulations.

**L225** – dual-origin "dominates" in the middle range is probably worth being explicit about. It looks to be the range should be 1-40 where dual-origin is the dominant category, not 1-10?

Corrected (actually the upper range is closer to 50 than 40).

**Fig 4** – "omit that panel" – but then you include the dashed contour on panel **e**. I would suggest you should be either including panel **j**, or omitting the dashed contour from panel e.

We now omit the dashed contour from panel e.

**L245** – "converging" – It is not clear to me that they are converging. Similar overlap of dashed and solid contours in both 3-4h and 7-8h. What is this statement based on?

This was based on a comparison of panels i) and e). However, this isn't fully correct, given that panel e) shows crystals that can be classified as "in situ" cirrus. Nevertheless, it is true that based on the panels shown, the difference between 3-4 and 7-8 h is small. It thus becomes harder to distinguish in situ from detrained ice crystals in all but the early development stage.

Over time, in situ cirrus tend to lose ice number and size, approaching microphysical properties that often overlap with those of aged anvil cirrus.

**Fig9** – Is the tailing off of both model an observation from the 5/3 slope robust or a result of limited data? If the latter, consider showing uncertainty on the plot or setting a threshold for what parts of the lines are robust and to be plotted.

Vertical wind spectra from models flatten out at the model effective resolution, which varies across the different GSRMs and SAM simulations shown in Figure 10. Aircraft observations are most suitable for investigating vertical wind variability < 10 km and flatten out between 10 and 100 km possibly due to a smaller signal-to-noise ratio, limited data, and/or changing atmospheric dynamics.

This was also added to the manuscript.

**L413-414 -** "We also perform a *short* sensitivity test…" What does short mean in this context? This explanation would be better suited in the methods.

The sensitivity test is as long as the other simulations (30 days), so "short" is removed.

**L415** – (not shown) Can it be added to the appendix?

We added it as an additional panel to Figure A1.

**Fig5** caption – It would be helpful to reference relevant text for each modification.

Done.

**Conclusions** – The first paragraph says about tracers first but then the following text describes model improvement results first. That's fine but it highlights that probably the model improvements results should come before tracer results in the main text too?

Technical corrections

**L313** – "Of particular significance are the high-frequency fluctuations". I'd change the order of the sentence: The high-frequency fluctuations are particularly significant...

Order switched to:
Of particular significance are the high-frequency fluctuations, which are approximately ten times larger than the slow synoptic scale motion (Atlas et al., 2023). Accurately capturing updraft variability is thus critical for modeling cirrus cloud formation and properties (Barahona et al., 2017).

**REVIEWER 2**

In "Tropical cirrus evolution in a km-scale model with improved ice microphysics", Gasparini et al. present new results on the lifecycle, microphysical and radiative properties of tropical cirrus. The authors employ Lagrangian methods to study these cloud properties, however unlike previous studies they introduce the novel application of passive, online tracers to track not only the lifetime of the cirrus but also link the measured properties to the formation process allowing deeper insights and comparisons into the relative contributions of anvil and in situ cirrus than previous studies. Comparisons between modelled cirrus properties, satellite retrievals and in situ aircraft observations are made both to provide cirrus cloud properties including their radiative impact, as well as to guide improvements to the model microphysics scheme to better match observed cirrus properties. The manuscript is well written and presented.

Comments:

Line 25: Is precipitation the main reason for the reduction is mass? Except for MCSs with a large stratiform component, I would expect sublimation to be the main sink of ice mass. In terms of optical thickness, is this not primarily driven by the divergence of the anvil cloud? I don't believe this reference is the most appropriate, or that a reference is necessary here.

For thick anvils, the impact of precipitation formation (or aggregation in the language of climate modeling) and sedimentation is thought to be greater than that of sublimation, especially in near-saturated conditions, which are common in the tropical upper troposphere. We updated the reference to Gasparini et al. (2021, doi: 10.1029/2020jd033487), who conducted a microphysical process rate analysis in Figure 9 for the detraining of tropical anvil air parcels. However, unlike their results, we expect sublimation to become the dominant ice sink in the case of thin anvils (several hours after detrainment).

Line 36: I don't fully agree with this statement (on the pros of passive tracers). The main trade off for passive tracers is that they have to be run online, meaning that this approach cannot be applied retrospectively to existing model runs, arguably meaning that they are less flexible. In addition, tracer advection is a large cost in many km-scale models. I would argue that the main benefits of passive tracers is that they are more accurate and can allow further insights into complex processes including cirrus origin, which could be the basis for a call for more modellers to include them in large model runs (e.g. DYAMOND) for community use.

This statement compares passive tracers to trajectories from a high-resolution modeling perspective.

It's difficult to envision applying trajectories to km-scale model output in an "offline" manner (one would require highly frequent model output for that). Therefore, trajectories must be run online with predetermined starting points. In comparison to that, we believe the use of passive tracers is simpler as it demands no apriori choices.

Although it is somewhat speculative since we cannot compare the two methods within the same modeling framework, we are convinced by the "simplicity" argument based on our previous experience with trajectories. We also believe passive tracers are computationally more efficient because they only need to advect a scalar field. In contrast, trajectories require interpolating wind fields for each particle, which becomes problematic when trying to obtain climatologically relevant results beyond limited case studies. Trajectories also likely require more memory than passive tracers.

Section 2.1.2: This subsection is nice, but could be improved by briefly outlining the issues with the standard P3 scheme at the start to provide context for the modifications.

Very briefly mentioned now.

Line 155: Is the lack of observations near convective cores accounted for when comparing with the model output later in the manuscript?

No, except that it's mentioned to be cautious about it. We don't expect this to affect the mean properties, as active convection only occurs in a small fraction of Earth and our model. However, it may lead to uncertainty in the ICNC-R_ice space results, particularly in the area with short "time after detrainment."

Additionally, the statement "the dataset contains very few measurements within or very close to active deep convection" has been revised to "limited measurements." See Fig. A3 for justification.

Line 181: While this approach provides a CRE estimate closer to the net CRE, it hides effects of differences in cirrus properties throughout the diurnal cycle. Comparing figure B2 to 10, in appears that the differences between CERES and model CREs are relatively larger when only sampling around 1:30pm. Could an alternative approach be to take compare both the night-time and daytime CERES overpasses? This could be included as a supplementary figure.

The CRE's bias is largely due to the overestimation of the SW CRE. This problem has been partially resolved with a bug fix, as described in the text: Additionally, a recently identified bug in the ice optics parameterization likely increased the optical depth and radiative effects per unit IWP by approximately 15%. However, this issue could not be addressed in the current simulations. The relative anomaly of the SW CRE between the regridded SAM model data and the CCCM is similar in Figures 4 (formerly Figure 10) and B2, ranging from 25 to 30% (see panels b in Figures 4 and B2).

Furthermore, based on our SAM model simulations, the difference in ice cloud frequency between the diurnal mean (Figure 4a) and the daytime-only data (Figure B2a) is small. We believe that other factors, such as the CCCM's inability to detect thin cirrus, are much more important than the diurnal cycle itself.

The diurnal cycle of anvil clouds has been studied using a similar version of the SAM model in the RCE modeling framework (Gasparini et al., 2022; doi:10.1175/JCLI-D-21-0211.1) as well as in observations (e.g., Wall et al., 2020; doi:10.1175/JCLI-D-20-0204.1). Therefore, we will not delve further into this topic in this work.

Line 210: To clarify, dual-original cirrus is defined as new ice nucleation occurring less than 30 hours after detrainment?

The threshold is now changed to 24 hours. Moreover, the time after nucleation must be shorter than the time after convection. As mentioned above, we also apply an additional condition to address issues related to ice crystal sedimentation. As described in the manuscript:

*Dual-origin cirrus are clouds detrained within the last 24 hours that have a shorter time since in situ nucleation compared to time after detrainment. These typically form within or near anvils or in cirrus clouds close to active convection (Fig.\ref{fig5-snapshots}d). To correct for classification biases due to ice crystal sedimentation impacting older anvils and in situ cirrus, clouds that were initially classified as in situ cirrus but located below dual-origin cirrus are reassigned to the dual-origin category.*

Figure 10: What is the lower limit of IWP measured by the different satellite products?

CloudSat and CALIPSO products (DARDAR and 2C-ICE) have been shown to detect clouds with an IWP of approximately 0.5 g m$^{-2}$ (Sourdeval et al., 2016, doi: 10.1002/qj.2889). Another study found that CALIPSO lidar reliably detects clouds up to an optical depth of approximately 0.01, which corresponds to an IWP of approximately 0.2 g/m² (Balmes & Fu, 2018, doi: 10.3390/atmos9110445).

The CCCM includes the IWP derived from MODIS, with a lower limit of about 4 g m$^{-2}$ (Sourdeval et al., 2016, doi: 10.1002/qj.2889). Because of that CCCM misses many thin, high clouds. Additionally, we note that there is no description of the detection limits of the MODIS retrieval algorithm used in the CCCM metadata, the CCCM user manual, or related publications.

We added the information on retrieval limits to the data section of the manuscript in Section 2.4.

Line 357: This result could be shown more clearly by including a figure showing the CRE x frequency contribution from each of detrained, in situ and dual-origin cirrus in the SAM model similar to fig. 3c, along with the total CRE contribution integrated over each distribution.

This comment was taken into account. We show results in the new Table 1 and describe them in the new Section 4.2.

Technical corrections:

Eq. 6/7: Order of clear sky and all sky flux terms is different between eq. 6 and 7, should be made consistent for clarity

Corrected.

Line 216: Missing reference to fig. 3a

Included.

Figure 10: Clarity of legends could be improved by including all labels in one legend for fig. 10a and explicitly showing all colour/pattern combinations for fig. 10b/c rather than colour for LW/SW/net and pattern for model/CCCM

On 10a (now Figure 4a): This is how we originally did the figure, but we think the current version is clearer.
On 10b/c (now Figure 4b/c): We prefer to avoid increasing the amount of ink and leave it as it is, as it is also intuitive to think about LW and SW and net flows (which are respectively strongly positive, strongly negative, and something in between).

Review of Title: Tropical cirrus evolution in a km-scale model with improved ice microphysics

Author(s): Blaž Gasparini et al.

General Comments

The authors present an analysis of temporal evolution of different cirrus types with the use of improved microphysical representation in the SAM model. A novel use of passive tracers in the model allows for tracking the cirrus evolution through part of the cirrus lifecycle and taking a look at the cloud radiative effects from these varied cirrus types. Model results are compared with satellite data and airborne observations collected in the Tropical Tropopause region with reasonable agreement, yet some discrepancies, which are discussed further in the text. The improvements in the microphysical model are a simple yet important step forward in representation of cirrus microphysics, and the passive tracers a unique tool to better track the influence of cloud formation mechanisms on characteristics and lifecycle. The concepts of the manuscript are fairly clear, but the text includes several general statements without direct quantification as well as general statements that point to another reference. The manuscript would benefit from further clarity through specificity, including brief summaries of the concept, definition, method, etc. referenced in the cited paper.

We followed the specific comments below and add some more specific information. Moreover (as mentioned in the beginning of this document), we changed the structure of the paper, which may help in improving its clarity.

Specific Comments

L120. Why is 80 minutes chosen for the arbitrary timescale? The reader is referenced to Gasparini et al. (2022), but for completeness it would be helpful to add a brief statement here.

It was selected as such to allow for analysis of processes that on timescales of hours to days. Also added in text on line 125: "*and tau_A = 80 minutes is an arbitrary decay timescale for A, that allows for analysis of processes on timescales of hours to days.*"

L148-150 and 381-383. Please elaborate on the bug that was recently fixed; is this bug present in the larger Kramer et al (2020) dataset? Is it also present in the original datasets in the NASA archives? On this latter point, somewhere in the paper or Code/Data Availability, reference to the original access for POSIDON, ATTREX, and CONTRAST data should be included.

The "bug" turned out not to be a bug after all, as described in the beginning of this document. We removed the connected text from the manuscript. The used POSIDON, ATTREX, and CONTRAST data are available at

https://b2share.fz-juelich.de/records/266ca2a41f4946ff97d874bfa458254c.

L155-156. Please quantify "few measurements?"

See the new Figure A3.

L158. Please clarify why the years 2007-2010 were chosen?  Would it be possible to utilize years that coincide with the aircraft data?

CloudSat radar is only available during the day as of April 2011, so we chose the only 4 full years with both day and night data for each of the datasets that use CloudSat retrievals. This includes DARDAR, 2C-ICE, and CCCM. Observations, on the other hand, show only snapshots sampling fewer clouds. In addition, CCCM data are only available from July 2006 to April 2011. Taking years with aircraft measurements (if they were available) would most likely make no difference to the satellite derived cloud properties and especially the ice water path.

L167. Please comment on any potential biases from utilizing nighttime-only data?

That will lead to some bias in the data, however, given that we examine a region fully covered by ocean, the diurnal cycle of high clouds is quite small and the bias due to the diurnal cycle will be much smaller compared to other model biases. A longer discussion on the diurnal cycle of tropical oceanic anvils is provided by , doi:10.1175/JCLI-D-20-0204.1 al., 2020 (observational perspective) and Gasparini et al., 2022 (modeling perspective, doi: 10.1175/JCLI-D-21-0211.1).

See also commented in text, section 2.4.1:
*" While this may lead to a bias, the diurnal cycle of tropical oceanic convection is small and of second order importance for the comparison with model results."*

Figure 4. How are these instances initialized?

With arrays filled with zeros, that can change value only if the selected conditions are true.

Figure 4. The overlayed contours of peak probability are helpful, thank you.

L243. "most of the in situ crystals are smaller than 30 um"  ..are you referring to the 7-8 h plot in Fig 4i?  For the other plots, there are still quite a number of ice crystals larger than 30 um.  Please clarify, or better yet, give numerical quantification to "most."

We refer to panels g-i (now Figure 7) and to in situ ice crystals in general, but particularly to the first several hours after nucleation. 62% of particles in panel g) are smaller than 30 μm compared to only 3% in panel b).

Changed to "since most of the in situ ice crystals are smaller than the freshly detrained ones..." (line 346) which is definitely the case.

Figure 5. "The number represents.." is rather unclear.  Would suggest being more specific: "The numbers in the upper left and upper right of the plot region.." Does this number have

a name? "Calculated separately for small and large particle sizes.." Please indicate the size range for each.

Considered in new Figure 2. This is called "2D total variation distance". As it is not a commonly used metric, we stick to a descriptive way of introducing it.

Figures 6 and 8. How many airborne samples are represented in each bin? Can you provide an indication of aircraft time spent sampling at the various temperature bins?

We added a figure showing that (Figure A3).

Appendices. Temperature is given as degrees C throughout the paper, then referenced in degrees Kelvin in appendices. Would suggest keeping consistency

Brightness temperature is almost always used in units of K, so we stick to K, despite agreeing with the reviewer on the consistency argument.

Technical Corrections

L86: "further refinements to would be" – remove "to"

Done

---

## Author Response (AR2)

**Response to reviewer's comments, Gasparini et al., 2025**

We thank the reviewer for their final comments.

In addition to the requested revisions, we have also added a new IWC estimate to Figure 3 (panels a and b), based on Clausius-Clapeyron relationship. This estimate is introduced in the main text (page 10, lines 236-241) and fully derived in Appendix C.

We are content with the article being published with the previous consideration of two minor corrections:
1- Both Fig.6 and Table 1 captions should be explicit that they are based on the simulated data. Even though it is made clear earlier in the section, the phrasing in the text referencing the figure/table is not explicit about this, so it would be good to remind the reader in the captions.

Done.

2- The adjustments to the in-situ/anvil illustration (Fig.4) are an improvement. However, I still take issue with it. On what basis do you decide what range of IWP the convective core spans? — this has not been part of your analysis. If I saw this in a presentation, I would take no issue, it's a pretty figure. But this is to be published, and anyone can view and consider it multiple times. Readers may have very different levels of scientific literacy or understanding of anvil clouds. Combining illustrations with plotted data risks encouraging assumptions based on nothing. Are there PhD students that will waste their time trying to confirm that the anvil core spans 4000+ g m-2 range of IWP? If you have based this on something, then I take it back, but perhaps express the reasoning in the caption. I have stated my, probably pedantic, point. If the authors and editor remain happy with the illustration, then please go ahead and publish.

We revised Figure 4 to better reflect the underlying data and added clarification regarding the deep convective core. Now it extends in the ice water path range between 1 and 10 kg/m$^2$, following the "deep core" definition by Sokol and Hartmann, 2020 (https://doi.org/10.1029/2020JD033107).

We share the reviewer's concern about the potential pitfalls of combining qualitative illustrations with quantitative data. We added a statement to the text (lines 260-261) to clarify that the IWP range reflects statistical occurrence, not the extent of an individual convective column. We hope this clarification reduces the risk of misinterpretation by less experienced readers.